# Rapid feedback on hospital onset SARS-CoV-2 infections combining epidemiological and sequencing data

Oliver Stirrup[1]*, Joseph Hughes[2], Matthew Parker[3,4,5], David G Partridge[6,7], James G Shepherd[2], James Blackstone[8], Francesc Coll[9], Alexander Keeley[6,7], Benjamin B Lindsey[6,7], Aleksandra Marek[10], Christine Peters[10], Joshua B Singer[2], The COVID-19 Genomics UK (COG-UK) consortium, Asif Tamuri[11], Thushan I de Silva[6,7], Emma C Thomson[2,12,13], Judith Breuer[14]*

[1]Institute for Global Health, University College London, London, United Kingdom; [2]MRC-University of Glasgow Centre for Virus Research, Glasgow, United Kingdom; [3]Sheffield Bioinformatics Core, The University of Sheffield, Sheffield, United Kingdom; [4]Sheffield Institute for Translational Neuroscience, The University of Sheffield, Sheffield, United Kingdom; [5]Sheffield Biomedical Research Centre, The University of Sheffield, Sheffield, United Kingdom; [6]Sheffield Teaching Hospitals NHS Foundation Trust, Sheffield, United Kingdom; [7]The Florey Institute for Host-Pathogen Interactions & Department of Infection, Immunity and Cardiovascular Disease, Medical School, University of Sheffield, Sheffield, United Kingdom; [8]The Comprehensive Clinical Trials Unit at UCL , University College London, London, United Kingdom; [9]Department of Infection Biology, Faculty of Infectious and Tropical Diseases, London School of Hygiene & Tropical Medicine, London, United Kingdom; [10]Clinical Microbiology, NHS Greater Glasgow and Clyde, Glasgow, United Kingdom; [11]Research Computing, University College London, London, United Kingdom; [12]Institute of Infection, Immunity and Inflammation, College of Medical, Veterinary and Life Sciences, University of Glasgow, Glasgow, United Kingdom; [13]Department of Infectious Diseases, Queen Elizabeth University Hospital, Glasgow, United Kingdom; [14]Division of Infection and Immunity, University College London, London, United Kingdom

*For correspondence:
oliver.stirrup@ucl.ac.uk (OS);
j.breuer@ucl.ac.uk (JB)

Competing interests: The authors declare that no competing interests exist.

## Abstract

**Background:** Rapid identification and investigation of healthcare-associated infections (HCAIs) is important for suppression of SARS-CoV-2, but the infection source for hospital onset COVID-19 infections (HOCIs) cannot always be readily identified based only on epidemiological data. Viral sequencing data provides additional information regarding potential transmission clusters, but the low mutation rate of SARS-CoV-2 can make interpretation using standard phylogenetic methods difficult.

**Methods:** We developed a novel statistical method and sequence reporting tool (SRT) that combines epidemiological and sequence data in order to provide a rapid assessment of the probability of HCAI among HOCI cases (defined as first positive test >48 hr following admission) and to identify infections that could plausibly constitute outbreak events. The method is designed for prospective use, but was validated using retrospective datasets from hospitals in Glasgow and Sheffield collected February–May 2020.

**Results:** We analysed data from 326 HOCIs. Among HOCIs with time from admission ≥8 days, the SRT algorithm identified close sequence matches from the same ward for 160/244 (65.6%) and in

the remainder 68/84 (81.0%) had at least one similar sequence elsewhere in the hospital, resulting in high estimated probabilities of within-ward and within-hospital transmission. For HOCIs with time from admission 3–7 days, the SRT probability of healthcare acquisition was >0.5 in 33/82 (40.2%).

**Conclusions:** The methodology developed can provide rapid feedback on HOCIs that could be useful for infection prevention and control teams, and warrants further prospective evaluation. The integration of epidemiological and sequence data is important given the low mutation rate of SARS-CoV-2 and its variable incubation period.

**Funding:** COG-UK HOCI funded by COG-UK consortium, supported by funding from UK Research and Innovation, National Institute of Health Research and Wellcome Sanger Institute.

## Introduction

Nosocomial transmission of SARS-CoV-2 presents a significant health risk to both vulnerable patients and to healthcare workers (HCWs) (*Kursumovic et al., 2020*; *Wang et al., 2020*; *Leclerc et al., 2020*; *The DELVE Initiative, 2020*; *Shah et al., 2020*). There is a variable incubation period, extending up to day 14 from exposure to the virus in symptomatic cases (*Lauer et al., 2020*). It is also known that transmission is possible from asymptomatic or presymptomatic carriers (*He et al., 2020*; *Rivett et al., 2020*; *Oran and Topol, 2020*; *Lucey et al., 2021*), complicating identification of hospital acquisition among hospital onset COVID-19 infections (HOCIs) and tracing of likely sources of infection.

There is now substantial evidence from retrospective studies that genome sequencing of epidemic viruses, together with standard infection prevention and control (IPC) practice, better excludes nosocomial transmissions and better identifies routes of transmission than IPC investigation alone (*Brown et al., 2019*; *Houldcroft et al., 2018*; *Roy et al., 2019*). The development of rapid sequencing methods capable of generating pathogen genomes within 24–48 hr has recently created the potential for clinical IPC decisions to be informed by genetic data in near-real time (*Meredith et al., 2020*). Although SARS-CoV-2 has a low mutation rate (*Fauver et al., 2020*), sufficient viral diversity exists for viral sequences to provide information regarding potential transmission clusters (*van Dorp et al., 2020*). However, phylogenetic methods alone cannot reliably identify linked infections, and the need for clinical teams to gather additional patient data presents challenges to the timely interpretation of SARS-CoV-2 sequence data.

To overcome these barriers, we have developed a sequence reporting tool (SRT) that integrates genomic and epidemiological data from HOCIs to rapidly identify closely matched sequences within the hospital and assign a probability estimate for nosocomial infection. The output report is designed for prospective use to reduce the delay from sequencing to impact on IPC practice. The work was conducted as part of the COVID-19 Genomics (COG) UK initiative, which sequences large numbers of SARS-CoV-2 viruses from hospitals and the community across the UK (*COVID-19 Genomics UK (COG-UK), 2020*). Here we describe the performance of the SRT using COG-UK sequence data for HOCI cases collected from Glasgow and Sheffield between February and May 2020 and explore how it may have provided additional useful information for IPC investigations.

## Materials and methods

The SRT methodology is applied to HOCI cases, defined here as inpatients with first positive SARS-CoV-2 test or symptom onset >48 hr after admission, without suspicion of COVID-19 at admission. The SRT algorithm returns an estimate of the probability that each HOCI acquired their infection post-admission within the hospital, with information provided on closely matching viral sequences from the ward location at sampling and wider hospital. Results for individual HOCIs are evaluated in relation to the IPC classification system recommended by Public Health England (PHE), based on interval from admission to positive test: 3–7 days post admission = indeterminate healthcare-associated infection (HCAI); 8–14 days post admission = probable HCAI; >14 days post admission = definite HCAI (*Public Health England, 2020*). We also applied the PHE definition of healthcare-associated COVID-19 outbreaks (*Public Health England, 2020*) (i.e. ≥2 cases associated with specific ward, with at least one being a probable or definite HCAI) to ward-level data, and for each

outbreak evaluated whether there was one or more distinct genetic cluster. This was determined by consecutive linkage of each HOCI into clusters using a two single-nucleotide polymorphism (SNP) threshold (with HOCIs assigned to a genetic cluster if a sequence match to any member). Sequences with <90% genomic coverage were excluded from all analyses.

## Data collection and processing

### Glasgow

During the first wave of SARS-CoV-2, the MRC-University of Glasgow Centre for Virus Research collected residual clinical samples from SARS-CoV-2-infected individuals following diagnosis at the West of Scotland Specialist Virology Centre. Samples were triaged for rapid sequencing using Oxford Nanopore Technologies (ONT) for suspected healthcare-related infections or Illumina sequencing in all other cases (details in Appendix 1).

### Sheffield

Residual clinical samples from SARS-CoV-2-positive cases diagnosed at Sheffield Teaching Hospitals NHS Foundation Trust were sequenced at the University of Sheffield using ARTIC network protocol (*ARTIC Network, 2020*) and ONT. Throughout the epidemic, members of the IPC team were notified by the laboratory and by clinical teams of positive results and reviewed relevant areas to ensure optimisation of practice and appropriate management of patients. Electronic reports were created contemporaneously, including an assessment as to whether suspected linked cases were present based on ward-level epidemiology. As part of SRT validation, these reports were accessed retrospectively by a study team member blind to the sequencing data and each included HOCI case was defined as being thought unlinked to other cases, a presumed index case in an outbreak or a presumed secondary case.

## HOCI classification algorithm

The sequence matching and probability score algorithm is run separately for each 'focus sequence' corresponding to a HOCI. We use associated metadata to assign other previously collected sequences to categories representing where the individual may be part of a SARS-COV-2 transmission network:

- Unit reference set: individual could be involved with transmission on same unit (ward/ICU, etc.) as focus sequence (look-back interval: 3 weeks)
- Institution reference set: individual could be involved with transmission in same institution/hospital as focus sequence (look-back interval: 3 weeks)
- Community reference set: individual could be involved with transmission outside of focus sequence institution (look-back interval: 6 weeks)

It is possible for samples to be members of multiple reference sets. For example, an outpatient may be involved in SARS-CoV-2 transmission at the institution they attended and/or in community transmission.

For each run of the algorithm, pairwise comparisons are conducted between the focus sequence and each sequence within the unit reference set, institution reference set and community reference set. A reference set sequence is considered a close match to the focus sequence if there is a maximum of two SNP differences between them. This choice was based on reported healthcare-associated outbreak events (*Meredith et al., 2020*; *Rockett et al., 2020*) and the overall mutation rate of SARS-CoV-2 (details in Appendix 1).

### Probability calculations

We use an expression of Bayes theorem to estimate probabilities for post-admission infection of each focus case divided by exposure on the unit, within the rest of the institution and from visitors (if allowed). An estimate of the prior probability ($P_{prior}$) of post-admission infection for each focus case is modified to a posterior probability according to the information provided by the sequence data. The algorithm is based on sound statistical principles, but involves heuristic approximations.

In symptomatic focus cases, we base $P_{prior}$ on the time interval ($t$) from admission to date of symptom onset or first positive test (if date of symptom onset not recorded). We calculate $P_{prior} = F$

(*t*), where *F*() is the cumulative distribution function of incubation times (*Lauer et al., 2020*) (derivation in Appendix 1).

In theory, it would be optimal to use all of the information in the *exact* sequences observed. However, with the goal of constructing a computationally simple algorithm, we base our calculations on the probability of observing a *similar* sequence (within two SNPs) to that actually observed for each focus case conditional on each potential infection source/location: infection in the community, current unit/ward or elsewhere in the hospital/institution, or from a visitor. For the unit and hospital, we estimate this probability using the observed sequence match proportion (on pairwise comparison to the focus sequence) in the unit reference set and institution reference set, respectively. For community- or visitor-acquired infection, we use a weighted proportion of matching sequences in the community reference set, with weightings determined by a calibration model that describes geographic clustering of similar sequences among community-acquired infections (described in Appendix 1). The geographic weighting model was fitted separately for each study site using sequences strongly thought to represent community-acquired infection: all community-sampled sequences and patients presenting to the Emergency Department with COVID-19, excluding those recorded as being HCWs.

## Software

The analysis was conducted in R (v. 4.0.2, R Foundation, Vienna) using sequence processing and comparison functions from *ape* (v. 5.4) and geospatial functions in the *PostcodesioR* (v. 0.1.1) and *gmt* packages (v. 2.0). R code to run the algorithm is available (*Stirrup, 2021*), and it has also been implemented as a standalone SRT for prospective use (*HOCI Sequence Reporting Tool working group, 2020*) within COV-GLUE (*Singer et al., 2020*).

# Results

## Study populations

### Glasgow

The Glasgow dataset included 1199 viral sequences (available as of 23 June 2020): 426 were derived from community sampling sites, 351 from patients presenting to Emergency Department or acute medical units, 398 from hospital inpatients and 24 from outpatients. Limited data were available regarding the total number of HCWs testing positive and their identification among community samples, but 15 sequences were recorded as being from HCWs. First positive test dates ranged from 3 March to 27 May 2020. All consensus sequences had genomic coverage >90%.

We applied the SRT algorithm to data from three hospitals with required metadata available, for which 128/246 inpatient cases with sequences were HOCIs. Two of these patients had been transferred from another hospital within 14 days prior to their positive test and were not processed as focus sequences. One inpatient without recorded sampling location was excluded, leaving 125 HOCIs for analysis. Population sequencing coverage was 536/1578 (34.0%) overall for patients at the three hospitals and 128/328 (39.0%) for HOCIs specifically (*Appendix 1—figure 1*).

### Sheffield

The Sheffield dataset included 1630 viral sequences with accompanying metadata (available as of 10 October 2020): 714 were from inpatients, 117 were from outpatients and 799 were from HCWs. For this retrospective evaluation, 447/714 inpatient samples taken on date of admission were assumed to represent community-onset cases and used to calibrate the model. First positive test dates ranged from 23 February to 30 May 2020. One sequence with genome coverage <90% was dropped from further analysis (an inpatient on date of admission). 201 of the inpatients were HOCIs. Population sequencing coverage was 714/977 (73.1%) overall for inpatients, 201/261 (77.0%) for HOCIs specifically and 799/962 (83.1%) for HCWs.

## Comparison to standard PHE classification

SRT algorithm results in comparison to standard PHE classifications are summarised in *Figure 1* and *Table 1*. The majority of HOCI cases in Glasgow (78/125, 62.4%) and over a third in Sheffield (71/201, 35.3%) met the definition of a definite HCAI and so are known to have acquired the virus post-

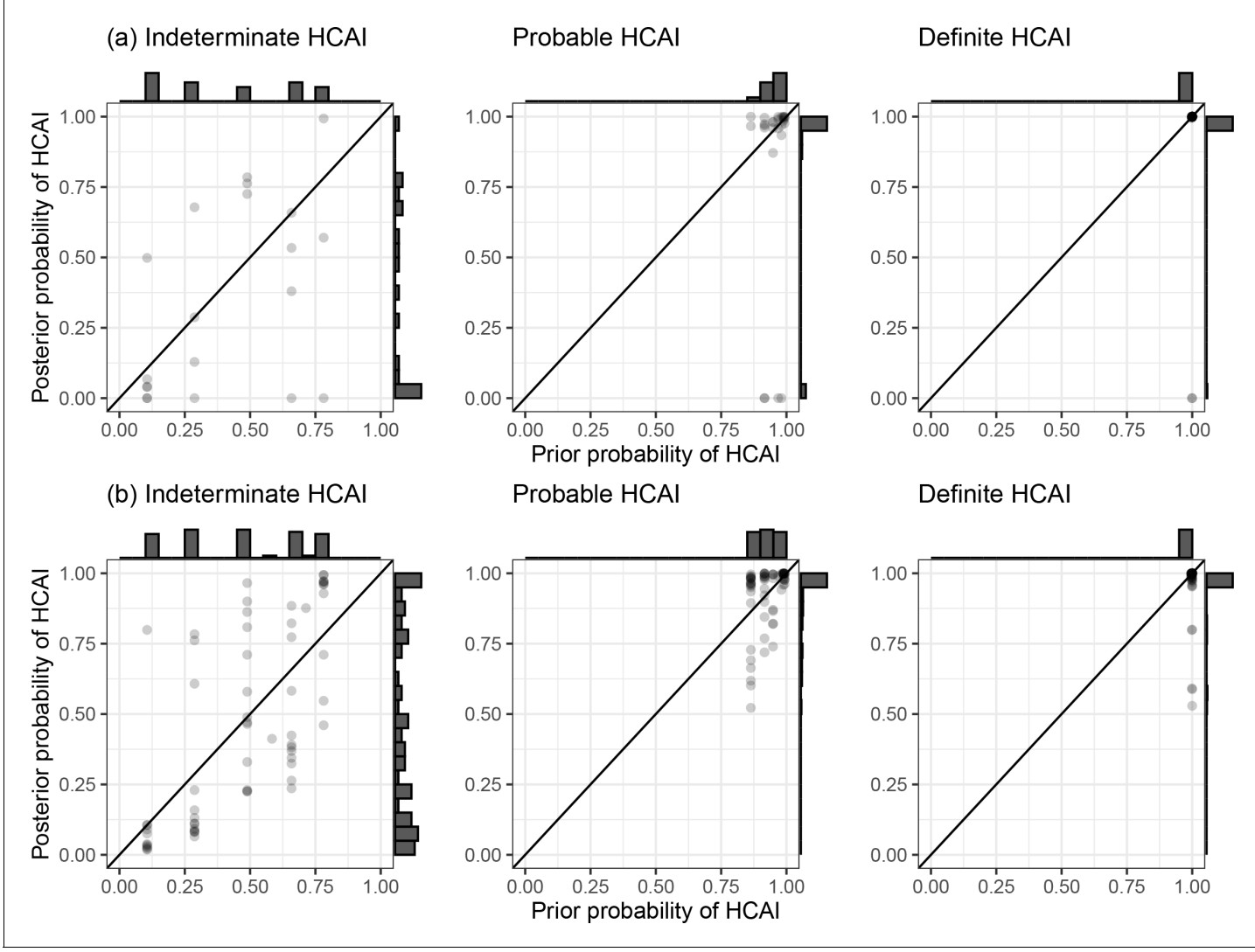

**Figure 1.** Plots of posterior probability of healthcare-associated infection (HCAI) against prior probability of HCAI. Plot of the posterior probability of healthcare-associated infection (HCAI) for (a) Glasgow and (b) Sheffield hospital onset COVID-19 infection cases from the sequence reporting tool algorithm against the prior probability of HCAI based only on time from admission to diagnosis, grouped by standard infection prevention and control classification recommended by Public Health England. Marginal histograms are displayed with bin-widths of 0.05.

admission irrespective of sequencing results. The probable HCAI cases formed the next largest group at each site. Overall, the SRT algorithm identified close sequence matches from the same ward for 66.4% of definite and 64.2% of probable HCAIs, indicating likely within-ward transmission (examples in case studies). When one or more close sequence match was identified on the focus sequence's ward, the SRT probability of infection on the ward was >0.5 in 185/189 cases (*Figure 2*). For indeterminate HCAIs, the SRT probability of HCAI was >0.5 in 33/82 (40.2%), and in 27/33 (81.8%) a close sequence match on the ward was present. Overall, 14/125 (11.2%) HOCIs in Glasgow and 175/201 (87.1%) in Sheffield had at least one close sequence match to a HCW sample, reflecting the much greater availability of sequences from HCWs in the Sheffield dataset.

In 16/244 (6.6%) cases that met the probable or definite HCAI definitions, there was no sequence match within the hospital; this is likely due to incomplete sequence data from SARS-CoV-2 hospitalised cases and staff (with population sequencing coverage <40% patients and very limited for staff from Glasgow and ≈75% of patients and staff in Sheffield) and the presence of asymptomatic and/ or undiagnosed carriers. To reflect this the SRT will report 'This is a probable/definite HCAI based on admission date, but we have not found genetic evidence of transmission within the hospital' in

**Table 1.** Summary of sequence reporting tool outputs for the Glasgow and Sheffield datasets, according to standard IPC definitions recommended by Public Health England regarding likelihood of HCAI.

| | Glasgow data | | | Sheffield data | | |
| | IPC classification | | | IPC classification | | |
| | Indeterminate HCAI | Probable HCAI | Definite HCAI | Indeterminate HCAI | Probable HCAI | Definite HCAI |
|---|---|---|---|---|---|---|
| *n* HOCI cases | 20 | 27 | 78 | 62 | 68 | 71 |
| Time from admission to sample*, days | 4.5 (3–6) | 11 (9-13) | 48 (26-83) | 5 (4–6) | 9 (8–13) | 22 (17–31) |
| *Summary of sequence matches returned for each HOCI case* | | | | | | |
| Close sequence match on ward | 5 (25.0) | 15 (55.6) | 53 (68.0) | 24 (38.7) | 46 (67.6) | 46 (64.8) |
| No close sequence match on ward, but match within hospital | 8 (40.0) | 7 (25.9) | 19 (24.4) | 34 (54.8) | 21 (30.9) | 21 (29.6) |
| No close sequence match anywhere within hospital | 7 (35.0) | 5 (18.5) | 6 (7.7) | 4 (6.5) | 1 (1.5) | 4 (5.6) |
| Close sequence match to one or more HCW | 1 (5.0) | 0 (0) | 13 (16.7) | 55 (88.7) | 61 (89.7) | 59 (83.1) |
| No close sequence match anywhere within dataset | 2 (10.0) | 1 (3.7) | 4 (5.1) | 4 (6.5) | 1 (1.5) | 4 (5.6) |
| *Probability calculations* | | | | | | |
| Prior probability of HCAI† | 0.39 (0.11–0.66) | 0.97 (0.92–0.99) | 1.00 (1.00–1.00) | 0.49 (0.29–0.66) | 0.92 (0.86–0.99) | 1.00 (1.00–1.00) |
| Posterior probability of HCAI‡ | 0.33 (0.02–0.67) | 0.98 (0.96–1.00) | 1.00 (1.00–1.00) | 0.40 (0.11–0.80) | 0.98 (0.93–1.00) | 1.00 (0.99–1.00) |
| *Posterior probability of HCAI‡ category* | | | | | | |
| Low (<30%) | 10 (50.0) | 4 (14.8) | 2 (2.6) | 25 (40.3) | 0 (0) | 0 (0) |
| Moderately low (≥30% and <50%) | 2 (10.0) | 0 (0) | 0 (0) | 12 (19.4) | 0 (0) | 0 (0) |
| Medium (≥50% and <70%) | 4 (20.0) | 0 (0) | 0 (0) | 4 (6.5) | 5 (7.4) | 3 (4.2) |
| High (≥70% and <85%) | 3 (15.0) | 0 (0) | 0 (0) | 8 (12.9) | 7 (10.3) | 2 (2.8) |
| Very high (≥85%) | 1 (5.0) | 23 (85.2) | 76 (97.4) | 13 (21.0) | 56 (82.4) | 66 (93.0) |

Data shown as median (interquartile range) or *n* (%).

*Or first +ve test where known.

†Based on time from admission.

‡From source on ward or within hospital.

HCAI: healthcare-associated infection; HOCI: hospital onset COVID-19 infection; HCW: healthcare worker; IPC: infection prevention and control.

such situations. There were 26 HOCIs in the Sheffield dataset for whom it was recorded that visitors were allowed on the ward at time of sampling. In three of these, the estimated probability of infection from a visitor was between 0.4 and 0.5 (all had ≥18 days from admission and no ward close sequence matches).

Within the Sheffield dataset, we identified six wards with two genetically distinct outbreak clusters (of two or more patients) and three wards with three distinct outbreaks (see Case study 2). Standard IPC assessment had classified each as a single outbreak. We also identified 10 and 44 HOCIs in the Glasgow and Sheffield datasets, respectively, with no apparent genetic linkage to other HOCI cases on the ward but who met the PHE definition of inclusion within an outbreak event (*Table 2*).

## Comparison to local IPC conclusions in Sheffield

Contemporaneous notes by IPC teams in Sheffield classified 18/201 HOCIs as the index case in outbreaks. IPC staff defined an index case as the first detected in an environment regardless of prior inpatient stay and, correspondingly, of these 14/18 were the first sequence on their ward and one was the second (the first 1 day earlier from a different bay on the ward was also recorded as an index case, and IPC staff deemed a ward outbreak with unclear index or possibly two index cases). Of the 18 index cases, 11 showed at least one subsequent close sequence match on the same ward (the two index cases on a single ward were not genetically similar, and for 1/18 there were no

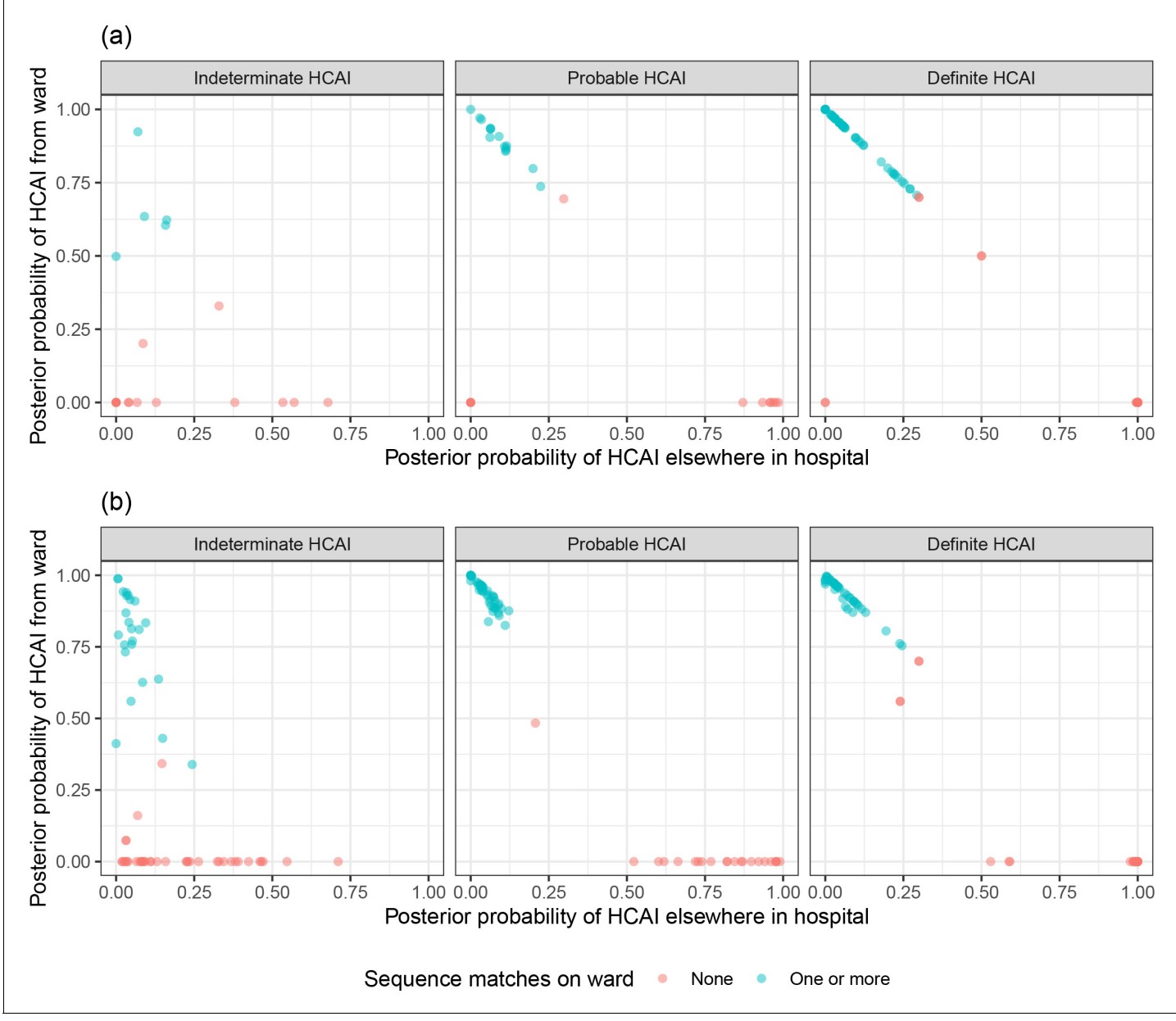

**Figure 2.** Plot of the posterior probabilities of healthcare-associated infection (HCAI) estimated using the sequence reporting tool algorithm from a source on the current ward versus a source elsewhere in the hospital for (a) Glasgow and (b) Sheffield hospital onset COVID-19 infection cases grouped by standard Public Health England classification. In cases where there are no close sequence matches in the dataset (including among community cases), the results returned are based solely on the priors and the metadata; this explains the fact that there are some cases with estimated posterior probability of infection on the ward greater than 0.5 for whom there were no sequence matches on the ward.

subsequent sequences from the ward). The median SRT probability of HCAI was 0.70 (IQR 0.22–1.00, range 0.04–1, >0.5 in 12/18).

A further 144/201 HOCIs were classified as being part of local outbreaks, and among these the median SRT probability of HCAI was 0.98 (IQR 0.89–1.00, range 0.02–1.00, >0.5 in 129/144) with one or more close sequence match on the same ward in 104/144. The remaining 39/201 HOCIs, including 10 that were not recorded as HOCIs at the time, were classified by the IPC teams as not being part of local outbreaks. Among these the median SRT probability of HCAI was 0.74 (IQR 0.23–0.99, range 0.02–1.00, >0.5 in 23/39), with one or more close sequence matches on the same ward in 7/39.

**Table 2.** Summary of distinct outbreak events for the Glasgow and Sheffield datasets, according to standard PHE definition and with the addition of sequence data.

| | Glasgow data | Sheffield data |
|---|---|---|
| *n* HOCI cases | 125 | 201 |
| *n* ward locations | 44 | 38 |
| | | |
| *Sequence matches per HOCI case* | | |
| *n* sequence matches from same ward, median (IQR, range) | 1 (0–5, 0–12) | 1 (0–4, 0–18) |
| *n* sequence matches from rest of hospital, median (IQR, range) | 3 (1–8, 0–52) | 27 (5–52, 0–150) |
| | | |
| *Standard PHE definition of outbreak event* | | |
| HOCI cases part of ward outbreak event, n (%) | 95 (76.0) | 184 (91.5) |
| *n* ward outbreak events | 17 | 24 |
| *n* HOCI cases per ward outbreak event, median (IQR, range) | 4 (2–8, 2–17) | 5 (3.5–10.5, 2–28) |
| Days from first to last case in outbreak, median (IQR, range) | 8 (6–15, 0–31) | 18 (13–34, 3–68) |
| *n* wards with more than one distinct outbreak event | 0 | 0 |
| | | |
| *Outbreak events with sequence linkage* | | |
| HOCI cases part of ward outbreak event, n (%) | 85 (68.0) | 140* (69.7) |
| *n* ward outbreak events | 16 | 33 |
| *n* HOCI cases per ward outbreak event, median (IQR, range) | 3.5 (2–8, 2–16) | 3 (2–4, 1–19) |
| Days from first to last case in outbreak, median (IQR, range) | 6 (4–9, 0–15) | 4 (2–8, 0–17) |
| *n* wards with more than one distinct outbreak event | 0 | 9† |

\* Includes two HOCIs which each showed a close sequence match to another case on the same ward with interval from admission to sample date ≤2 days.

†In three wards, there were three genetically distinct outbreak events.

HOCI: hospital onset COVID-19 infection; IQR: interquartile range; PHE: Public Health England.

## Case study 1

*Figure 3* shows a phylogenetic tree of eight HOCIs within a single ward at a Glasgow hospital (Hospital 5, Unit 93), alongside associated metadata and SRT probability outputs. The first HOCI detected (UID0032) was transferred from another hospital within the previous 2 weeks and so SRT output was not generated. All subsequent HOCIs return close sequence matches to at least one prior case on the ward, leading to SRT probability estimates of ward-acquired infection >0.9, even for UID0017 (an indeterminate HCAI). The phylogenetic tree indicates UID0032 has an SNP lacked by most of the cases identified on the ward, and therefore did not seed all of the cases in the outbreak cluster. Also shown is a single HOCI from a different ward in the same hospital (UID0025); this individual was an indeterminate HCAI, but a higher proportion of similar viral sequences within the hospital in comparison to their local community led to a SRT result of probable hospital-acquired infection.

## Case study 2

*Figure 4* shows phylogenetic trees relating to three distinct viral lineages identified on a single ward in the Sheffield dataset (classified by contemporaneous IPC investigation as a single outbreak). Two of these lineages also include sequences from inpatients sampled from other wards within the same hospital. Detailed ward movement data highlighted additional possible links between patients in the B.2.1 cluster. Both UID0149 and UID0157 were present at LOC0111 prior to their sample dates.

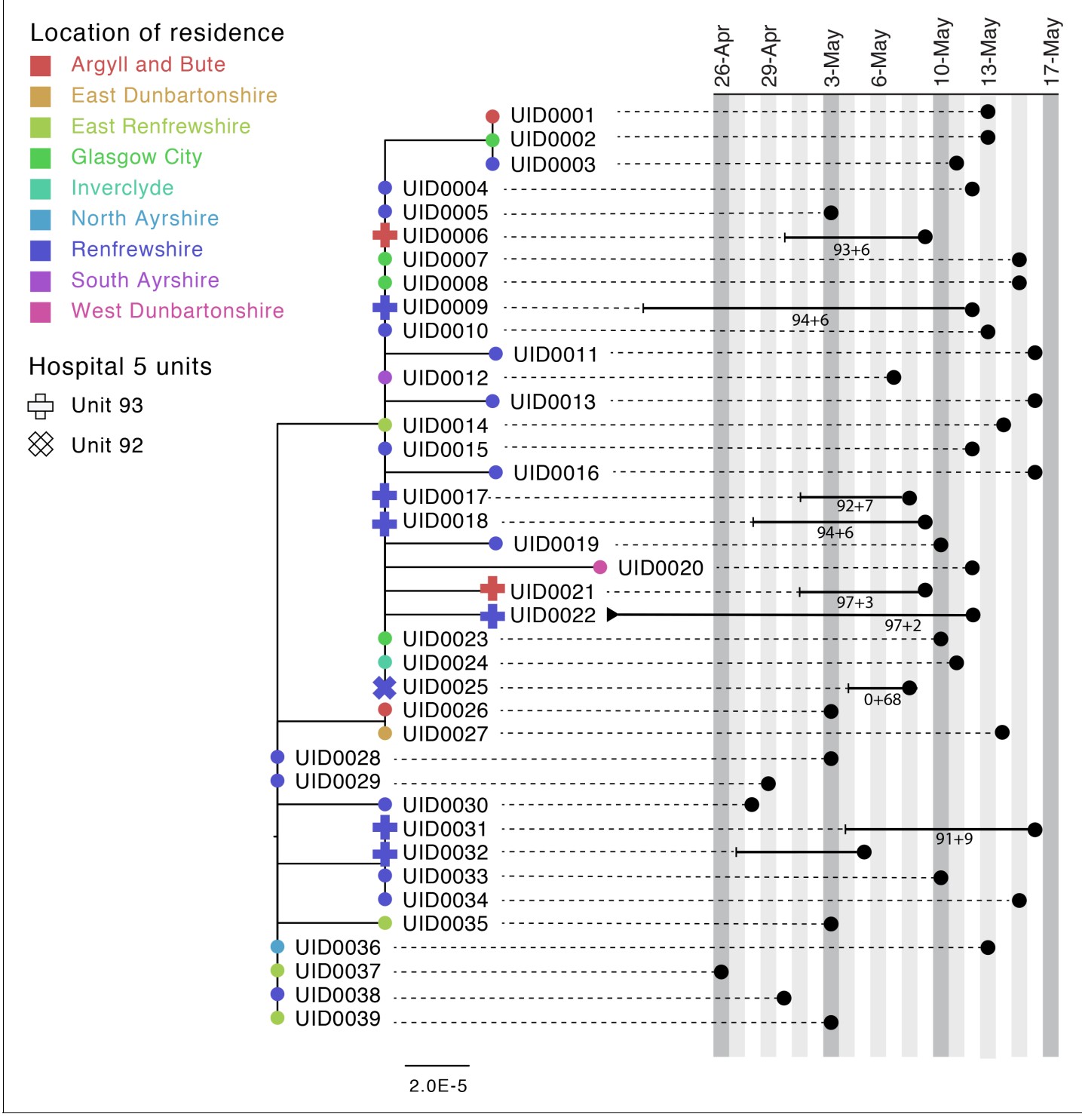

**Figure 3.** Maximum-likelihood phylogeny of the sequences found in Hospital 5 Unit 93 and Unit 92 up until the 16th of May of the Glasgow dataset. The black lines represent the time from admission to sampling. The values below the line are the posterior probability for unit infection + the posterior probability of hospital infection from the sequence reporting tool. The tip nodes are coloured according to the local authority area of the community surveillance sequences (circles) or of the patients (crosses).

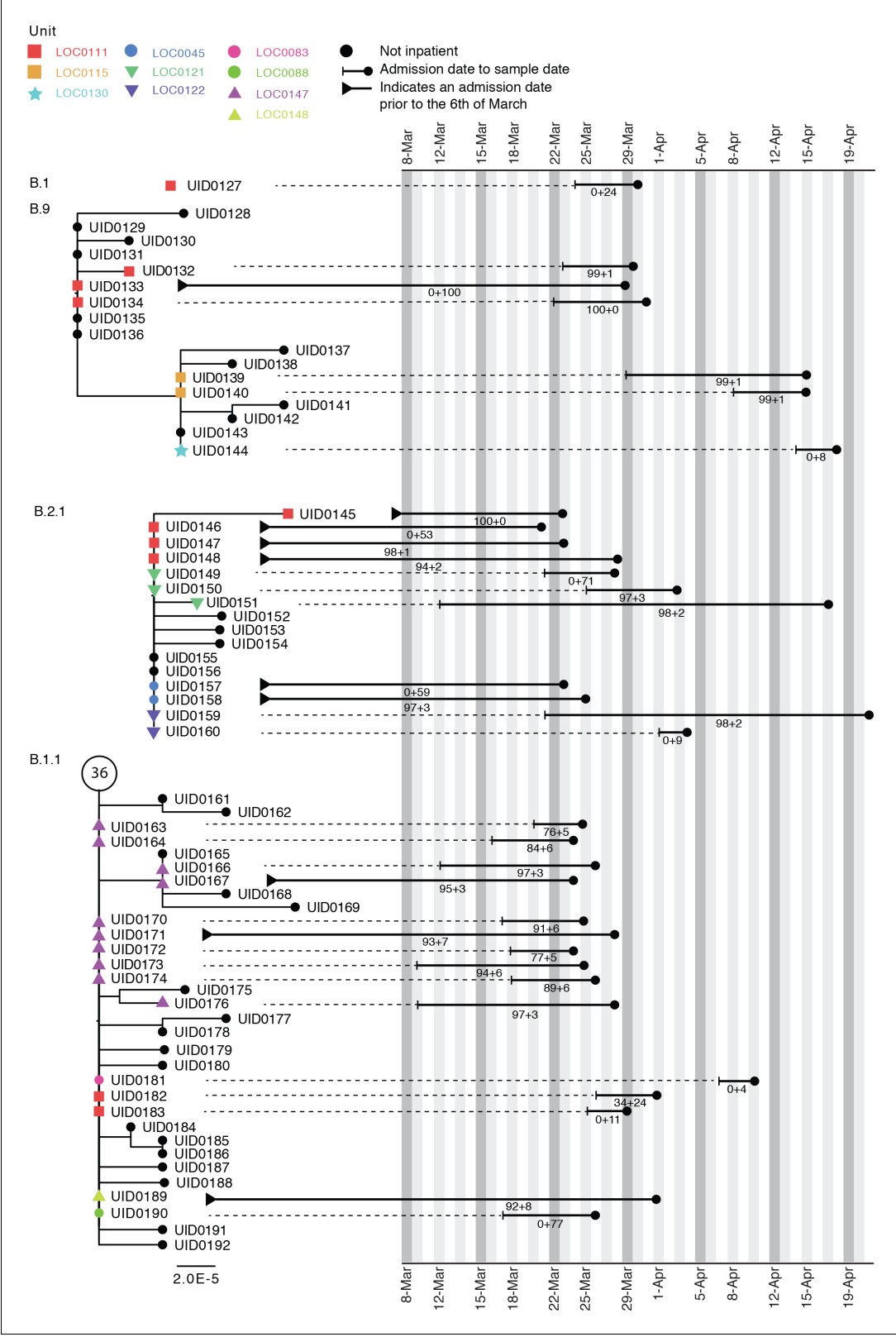

**Figure 4.** Maximum-likelihood phylogeny of the sequences found in location '0111' in the Sheffield dataset, also including patients at several other ward locations. The tree tip nodes are coloured according to ward locations. The black lines represent the time from admission to sampling. The values below the line are the posterior probability for unit infection + the posterior probability of hospital infection from the sequence reporting tool. The circle containing a number represents community sequences that are identical and at the base of this lineage (n = 36).

## Discussion

We have developed a novel approach for identification and investigation of hospital-acquired SARS-CoV-2 infections combining epidemiological and sequencing data, designed to provide rapid and concise feedback to IPC teams working to prevent nosocomial transmission. Through retrospective application to clinical datasets, we have demonstrated that the methodology is able to provide confirmatory evidence for most PHE-defined definite and probable HCAIs and provide further information regarding indeterminate HCAIs. Thus the SRT may allow IPC teams to optimise their use of resources on areas with likely nosocomial acquisition events.

While the SRT is not likely to change IPC conclusions in cases meeting the definition of 'definite' or 'probable' HCAI based on interval from admission to symptom onset, in 91% of cases it did identify patients in the same ward or elsewhere in the hospital who could plausibly be linked to the HOCI within a single outbreak event. Those definite and probable HOCIs without close sequence matches are likely to reflect transmission from sources within the hospital that have either not been diagnosed or who were diagnosed without viral sequencing. In such cases, it is impossible to calculate a probability of transmission and the SRT will simply state that no sequence matches were found within the hospital.

For cases meeting the definition of 'indeterminate healthcare associated', the probability scores returned would be useful for IPC teams. These probabilities are dependent on comparison to sequences from cases of community-acquired infection obtained either from direct community sampling or from patients sampled at admission. The Sheffield dataset was lacking the former data source, but the SRT nonetheless classified a similar proportion of 'indeterminate healthcare associated' HOCIs as community-acquired infections to that found in the Glasgow dataset (approximately 60%).

Current PHE guidelines define healthcare-associated COVID-19 outbreaks as two or more cases associated with a specific setting (e.g. ward), with at least one case having illness onset after 8 days of admission (*Public Health England, 2020*). However, the guidelines note that 'investigations of healthcare-associated SARS-CoV-2 infection should also take into account COVID-19 cases categorised as "indeterminate healthcare associated"' (i.e. onset 3–7 days after admission), for which our SRT output would be useful. In most HOCIs meeting this definition of inclusion within an outbreak event, we found evidence of clusters of similar viral sequences located on the ward concerned, and the SRT results were in line with available local IPC classifications in the majority of cases. However, a substantial minority (54/279) of HOCIs, although assumed to be part of a ward outbreak, were, in fact, isolated cases for which the sequencing data refuted genetic linkage to other sequences from the ward. The SRT also provided evidence of wards where IPC-defined outbreak events comprised two or three clearly distinct viral lineages.

The retrospective datasets analysed in this study represent the first few months of the COVID-19 epidemic in the UK, and nosocomial transmission of the virus in the UK during this period has previously been reported at multiple sites (*Meredith et al., 2020*; *Rickman et al., 2021*; *Carter et al., 2020*). HCWs were at increased risk of infection and adverse health outcomes (*Kursumovic et al., 2020*; *Wang et al., 2020*; *The DELVE Initiative, 2020*; *Shah et al., 2020*; *Houlihan et al., 2020*) and could have been important drivers of nosocomial transmission (*Rivett et al., 2020*). Data were limited for Glasgow, but the Sheffield dataset contained a large number of sequences obtained from HCWs, with population sequencing coverage for this group >80%, and there was a close sequence match to at least one HCW observed for 87% of HOCIs. Our analysis has not evaluated direction of transmission to or from HCWs, but they were clearly linked into transmission networks within the hospital. A limitation of the current SRT approach and of the retrospective data available is that they do not include detailed information regarding work locations for HCWs. However, prospective use of the SRT would allow IPC teams to investigate linkage from a HOCI to any HCWs flagged as having a close sequence match.

While a phylogenetic approach is useful in excluding direct transmission between cases, it can be more problematic to confirm transmission source (*Volz and Frost, 2013*). Phylogenetic models can evaluate the full genetic information provided by viral sequence data, but there are challenges in incorporating and summarising associated patient metadata in a timely fashion (*Villabona-Arenas et al., 2020*). The challenge of timely collection and standardisation of patient metadata is also relevant for use of the SRT that we have developed, but it is possible to automate such

processes through electronic patient record systems. There have been advances in recent years in the computational efficiency and workflow standardisation possible for phylogenetic analyses that have made it easier to use these methods for real-time investigation of outbreaks, for example, through the development of the Nextstrain project (*Hadfield et al., 2018*; *Huddleston et al., 2021*). However, there does not currently exist phylogenetic software for SARS-CoV-2 that produces reports or other outputs designed for direct and immediate use by IPC professionals. There will be cases in which phylogenetic analysis would provide information beyond that returned by the SRT, and the two approaches may be complementary to one another for outbreak investigation.

Comparison of SRT output to phylogenetic trees in a number of test cases suggested that some clusters of genetically similar cases identified within a specific ward likely represented more than one transmission event onto the ward from similar viral lineages circulating within the healthcare system. Whilst monophyletic clusters associated with a single location are easier to interpret, we consider the presence of viruses within a ward or hospital that are genetically similar to a HOCI as evidence for nosocomial infection even when they are not plausible transmission sources themselves, given the potential for asymptomatic transmission (*He et al., 2020*; *Rivett et al., 2020*; *Oran and Topol, 2020*; *Lucey et al., 2021*) and complex transmission networks (*Meredith et al., 2020*).

The SRT uses a number of heuristic approximations in order to provide an integrated summary of epidemiological and sequence data. However, this choice is associated with the limitation that it does not provide a full probabilistic model of potential transmission networks. Further development of the SRT would also aim to more fully incorporate patient movement data and shift locations for HCWs.

We believe that collaboration between methodologists, virologists, IPC clinicians and software engineers is essential in order to create workflows and reporting systems that will enable the routine use of pathogen sequence data for IPC. The SRT represents such a collaboration, and it has been designed to enable automation of the linkage and processing of viral sequence and patient meta-data and subsequent feedback of relevant information to IPC staff. The automated feedback provided by the SRT is nonetheless dependent on timely sequencing of a high proportion of viral samples from cases within the hospital concerned, ideally in combination with sequences also available from community-sampled cases. In the UK this has been possible through the national COG-UK project (*COVID-19 Genomics UK (COG-UK), 2020*). Denmark has also implemented high population-coverage sequencing of SARS-CoV-2 (*Bager et al., 2021*), but this is not the case for most countries. The emergence and rapid dominance of lineage B.1.1.7 in the UK (*Volz et al., 2021*) has provided a case study for the impact of national-level genomic surveillance, but further evidence is required to determine whether rapid sequencing is worth the necessary investment for routine use within IPC practice. This judgement would also be dependent on the available health infrastructure and resources at both the local and national levels.

Prospective evaluation of the SRT is currently underway within a multicentre study in the UK (*Blackstone et al., 2021*). This study and its accompanying research programme will evaluate the impact of routine viral sequencing and use of the SRT on IPC knowledge, actions and outcomes, and will include quantitative, qualitative (*Flowers et al., 2021*) and health economic analyses to help guide the future development of pathogen genomics for IPC.

Our novel approach to the investigation of HOCIs has shown promising characteristics on retrospective application to two clinical datasets. The SRT described allows rapid feedback on HOCIs that integrates epidemiological and sequencing data to generate a simplified report at the time that sequence data become available. Prospective evaluation is required in order to recommend use of the SRT in clinical practice, and this work is ongoing. The methodology has been developed for hospital inpatients, but the principles may also be applicable to other settings.

## Acknowledgements

COG-UK HOCI is funded by the COG-UK consortium, which is supported by funding from the Medical Research Council (MRC) part of UK Research and Innovation (UKRI), the National Institute of Health Research (NIHR) and Genome Research Limited, operating as the Wellcome Sanger Institute. JBr receives funding from the NIHR ULC/UCLH Biomedical Research Centre. FC is funded by Wellcome (grant number: 201344/Z/16/Z). MDP is funded by the NIHR Sheffield Biomedical Research Centre (BRC – IS-BRC-1215–20017). We acknowledge the help of the UCL Comprehensive Clinical

Trials Unit. We thank the NHS Greater Glasgow and Clyde and Sheffield Teaching Hospitals NHS Foundation Trust infection prevention and control teams for provision of data. We thank Michael Chapman for his assistance in the development of this project.

## Additional information

### Funding

| Funder | Grant reference number | Author |
|---|---|---|
| Wellcome | 201344/Z/16/Z | Francesc Coll |
| NIHR | | Judith Breuer |
| NIHR | IS-BRC-1215–20017 | Matthew Parker |
| Medical Research Council | | The COVID-19 Genomics UK (COG-UK) consortium |
| NIHR | | The COVID-19 Genomics UK (COG-UK) consortium |
| Wellcome | | The COVID-19 Genomics UK (COG-UK) consortium |

The funders had no role in study design, data collection and interpretation, or the decision to submit the work for publication.

### Author contributions

Oliver Stirrup, Software, Formal analysis, Investigation, Methodology, Writing - original draft, Writing - review and editing; Joseph Hughes, Data curation, Formal analysis, Investigation, Visualization, Writing - review and editing; Matthew Parker, David G Partridge, James G Shepherd, Alexander Keeley, Benjamin B Lindsey, Aleksandra Marek, Christine Peters, Data curation, Investigation, Writing - review and editing; James Blackstone, Francesc Coll, Project administration, Writing - review and editing; Joshua B Singer, Asif Tamuri, Software, Methodology, Writing - review and editing; Thushan I de Silva, Emma C Thomson, Conceptualization, Data curation, Supervision, Investigation, Writing - review and editing; Judith Breuer, Conceptualization, Supervision, Writing - original draft, Writing - review and editing

### Author ORCIDs

Oliver Stirrup (iD) https://orcid.org/0000-0002-8705-3281
David G Partridge (iD) http://orcid.org/0000-0002-0417-2016
Benjamin B Lindsey (iD) http://orcid.org/0000-0003-4227-2592
Judith Breuer (iD) http://orcid.org/0000-0001-8246-0534

### Ethics

Human subjects: Research Ethics for COG-UK Consortium and research undertaken under its auspices was granted by the PHE Research Ethics and Governance group as part of the emergency response to COVID-19 (24 April 2020, REF: R&D NR0195) and by the relevant Scottish biorepository authorities (16/WS/0207NHS and 10/S1402/33). This was a retrospective analysis on fully anonymized data, the collection of which did not involve any active research intervention. Consent therefore was neither required nor requested from individual patients.

### Decision letter and Author response

Decision letter https://doi.org/10.7554/eLife.65828.sa1
Author response https://doi.org/10.7554/eLife.65828.sa2

# Additional files

## Supplementary files

• Supplementary file 1. Comma separated value file containing a list of the COG-UK identification codes for viral sequences included in the analysis.

• Supplementary file 2. Details of the COVID-19 Genomics UK (COG-UK) consortium.

• Transparent reporting form

## Data availability

The sequence data analysed are included within publicly available datasets (https://www.cogconsortium.uk/data/), and a list of the relevant sequence identification codes is provided (Supplementary File 1). Due to data governance restrictions related to individual patient data linked to genetic sequences it is not possible to publicly share the associated meta-data. Requests for access to the data can be made by submission of a research proposal to the COG-UK Steering Committee (contact@cogconsortium.uk).

The following previously published dataset was used:

| Author(s) | Year | Dataset title | Dataset URL | Database and Identifier |
|---|---|---|---|---|
| The COVID-19 Genomics UK (COG-UK) consortium | 2020 | UK SARS-CoV-2 sequence data | https://www.cogconsortium.uk/data/ | Trimmed and masked alignment, Trimmed and masked alignment |

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

## Appendix 1

### Methods
#### Details of sequencing protocols
##### Glasgow

Sequencing with ONT followed the protocols developed by the ARTIC network (v1 and v2) https://artic.network/ncov-2019. The reads were aligned to the reference strain (MN908947) using mini-map2 (https://doi.org/10.1093/bioinformatics/bty191) and denoised using nanopolish (https://www.nature.com/articles/nmeth.3444) prior to primer trimming and consensus calling with iVar using a minimum depth of 20 reads (https://doi.org/10.1186/s13059-018-1618-7). Sequencing with Illumina also used the ARTIC network protocol for amplicon generation but was followed by a DNA KAPA library preparation kit (Roche) and indexing with NEBNext multiplex oligos (NEB) using seven PCR cycles. Libraries were pooled and loaded on a MiSeqV2 cartridge. Illumina reads were processed with the PrimalAlign pipeline (*Orton, 2021*). Briefly, reads were trimmed using trim_galore (https://www.bioinformatics.babraham.ac.uk/projects/trim_galore/) aligned to the reference using BWA (10.1093/bioinformatics/btp698). Then, amplicon primers were removed and the consensus called with a read depth of 10 using iVar (https://doi.org/10.1186/s13059-018-1618-7). Metadata associated with each sample was collated in a redcap database (https://www.project-redcap.org/).

##### Sheffield

Sequencing with ONT followed the protocols developed by the ARTIC network (v1 and v2) (https://artic.network/ncov-2019). Following base calling, data were demultiplexed using ONT Guppy using a high-accuracy model. Reads were filtered based on quality and length (400–700 bp), then mapped to the Wuhan reference genome and primer sites trimmed. Reads were then downsampled to $200\times$ coverage in each direction. Variants were called using nanopolish (*Simpson, 2021*) and used to determine changes from the reference. Consensus sequences were constructed using reference and variants called.

### Further details of reference set definitions
#### Data sources for algorithm

There are two potential sources of data for the HOCI classification algorithm. Firstly, there are *institution-sampled sequences*: these include all viral sequences from samples obtained within the institution/hospital. These sequences are linked to metadata providing basic information regarding the patient concerned and details of the sample from which the sequence was obtained. Secondly, there are *community-sampled sequences*: these include all relevant sequences obtained from samples from testing within the local community. These sequences are associated with a more limited set of linked metadata describing date of sample, residential outer postcode of subject and place of work if they are recorded as being a HCW.

#### Unit reference set

This dataset comprises all institution sequences sampled on or $\leq$3 weeks prior to (or $\leq$2 days after for the prospective version of the SRT) the sample date of the focus sequence and for which both the institution and the unit are the same as that for the focus sequence.

#### Institution reference set

This dataset comprises firstly all institution-sampled sequences from HCWs, outpatients and inpatients diagnosed >48 hr after admission for which the institution matches that of the focus sequence sampled on or $\leq$3 weeks prior to (or $\leq$2 days after for the prospective version of the SRT) the sample date of the focus sequence and for which the unit is either not the same as that for the focus sequence or is missing. Secondly, the dataset includes all institution-sampled sequences from A and E patients or inpatients diagnosed $\leq$2 days after admission for which the institutionID matches that of the focus sequence sampled between (inclusively) 3 weeks and 3 days prior to the sample date of

the focus sequence and for which the unit is either not the same as that for the focus sequence or is missing. Thirdly, this dataset also includes the subset of community-sampled sequences of HCWs at the same institution as the focus sequence.

### Community reference set

This dataset comprises firstly all community-sampled sequences sampled on or ≤6 weeks prior to (or ≤2 days after for the prospective version of the SRT) the sample date of the focus sequence. This dataset also includes institution-sampled sequences sampled on or ≤6 weeks prior to (or ≤2 days after for the prospective version of the SRT) the sample date of the focus sequence from all non-inpatient samples, and those inpatients for whom sample date and symptom onset date (if recorded) are both ≤2 days after the admission date.

Note that some institution-sampled sequences will contribute to both the community reference set and either the unit reference set or the institution reference set (e.g. outpatients sampled within 3 weeks prior to the focus sequence would be included in both the community reference set and the institution reference set). HCWs recorded among the community-sampled sequences within ≤3 weeks prior to the sample date of the focus sequence will also be included in both the community reference set and the institution reference set if their workplace matches the institution of the focus sequence.

## Formulae for probability calculations

Posterior of unit-acquired infection (UI) =

$$\frac{P_{prior}*P_u*P(seq\pm2SNPs|UI)}{P_{prior}*P_u*P(seq\pm2SNPs|UI)+P_{prior}*P_v*P(seq\pm2SNPs|VI)+P_{prior}*(1-P_u-P_v)*P(seq\pm2SNPs|II)+\left(1-P_{prior}\right)*P(seq\pm2SNPs|CI)}$$

Posterior of institution-acquired infection (II) =

$$\frac{P_{prior}*(1-P_u-P_v)*P(seq\pm2SNPs|II)}{P_{prior}*P_u*P(seq\pm2SNPs|UI)+P_{prior}*P_v*P(seq\pm2SNPs|VI)+P_{prior}*(1-P_u-P_v)*P(seq\pm2SNPs|II)+\left(1-P_{prior}\right)*P(seq\pm2SNPs|CI)}$$

Posterior of visitor-acquired infection (VI) =

$$\frac{P_{prior}*P_v*P(seq\pm2SNPs|VI)}{P_{prior}*P_u*P(seq\pm2SNPs|UI)+P_{prior}*P_v*P(seq\pm2SNPs|VI)+P_{prior}*(1-P_u-P_v)*P(seq\pm2SNPs|II)+\left(1-P_{prior}\right)*P(seq\pm2SNPs|CI)}$$

Posterior of community-acquired infection (CI) =

$$\frac{\left(1-P_{prior}\right)*P(seq\pm2SNPs|CI)}{P_{prior}*P_u*P(seq\pm2SNPs|UI)+P_{prior}*P_v*P(seq\pm2SNPs|VI)+P_{prior}*(1-P_u-P_v)*P(seq\pm2SNPs|II)+\left(1-P_{prior}\right)*P(seq\pm2SNPs|CI)}$$

With terms defined as follows:

$P_{prior}$: prior probability of post-admission infection for each focus case, based on time interval from admission to date of symptom onset or first positive test
$P_u$: prior probability of UI given post-admission infection (set based on expert opinion)
$P_v$: prior probability of VI given post-admission infection (set based on expert opinion)
$P(seq \pm 2SNPs|infectionsource/location)$: probability of observing a *similar* sequence (within two SNPs) to that actually observed for each focus case conditional on each potential infection source/location (estimated from sequence reference sets)

When there is a close sequence match found in any of the defined reference sets, the posterior probability estimates for UI, II, VI and CI will always sum to 1. However, when there is no close sequence match in any of the reference sets the posterior probability calculations are not valid and the algorithm will return the prior probabilities for each potential source/location of infection.

## Further details regarding sequence matching process

The ±2 SNP threshold for a close sequence match was initially based on reports of healthcare-associated outbreak events for which this was the maximum pairwise difference within clusters (Meredith: DOI:10.1101/2020.05.08.20095687 and Rockett: DOI: 10.1101/2020.04.19.048751). The outbreak events described included sequences with up to around 3 weeks between first and last samples. This SNP threshold is also supported by calculations using the overall mutation rate of SARS-CoV-2. If we

take the average mutation rate of the virus to be 24 SNPs/year (Nextstrain value 24 June, https://nextstrain.org/ncov/global?l=clock), then assuming independent (Poisson distributed) mutation events, ignoring the chance of mutations occurring at the same position in the genome and using a fixed generation time of 5 days, then there is an approximate:

> 72% chance of no new SNPs per generation
> 24% chance of one new SNP per generation
> 4% chance of two new SNPs per generation
> 0.4% chance of three new SNPs per generation

A two SNP threshold would therefore be expected to identify close sequence matches between direct transmission pairs in a large majority of cases. Ambiguous nucleotide positions will be considered to match if there is an overlap in the possible values for the two sequences. 'N' values recorded in either the focus sequence or comparison sequence will be considered to be a match at that position.

## Further details of prior probability calculations for post-admission infection

We calculate $P_{prior} = F(t)$, where $F()$ is the cumulative distribution function of a published log-normal distribution for incubation times (Lauer et al: doi:10.7326/M20-0504; μ = 1.621, σ = 0.418). For symptomatic HOCI cases, the IPC classifications recommended by PHE translate into the following value ranges for $P_{prior}$:

- indeterminate HCAI: 0.11 (onset 3 days post-admission) to 0.78 (onset 7 days post-admission)
- probable HCAI: 0.86 (onset 8 days post-admission) to 0.99 (onset 14 days post-admission)
- definite HCAI: $P_{prior} \geq 0.995$

For asymptomatic focus cases, we define our prior on the basis that some proportion of the cases detected will never become symptomatic ($P_a$) with the remainder going on to develop symptoms within the next few days (1 - $P_a$). We then define our prior probability of post-admission infection in these cases as

$$P_{prior} = (1 - P_a) * F(t + c) + P_a * F(t)$$

where $t$ is the interval from admissionDate to sampleDate, and $c$ is a constant reflecting the average interval within which we expect symptoms to appear (among those cases in which they do). $P_a$ is set at 0.4 based on the findings of a published review article (Oran and Topol: 10.7326/M20-3012), and $c$ is set to three based on a combination of expert opinion of the study PIs, the known distribution of time from infection to symptom onset and expert experience of asymptomatic screening.

### Source given post-admission infection

The model requires prior values for the probability of UI and VI given post-admission infection: $P_u$ and $P_v$, respectively. However, in specifying the model we define $P_u'$ as the probability of UI given post-admission infection when there are no visitors allowed on the ward, in which case the probability of VI is 0 and $P_v' = 0$. If visitors are allowed on the ward for the focus case, then we set $P_u = P_u' \times (1 - P_v)$.

Based on expert opinion of the clinical co-authors, $P_u'$ is set to different values according to the unit/ward type of the focus sequence with single-bed wards having a lower prior probability of unit post-admission infection than bay wards: 0.5 for single-bed wards and 0.7 for bay wards. We assumed a $P_v$ of 0.2. The $P_u$ values (when visitors are allowed) are therefore 0.4 for single-bed wards and 0.56 for bay wards. The largest of the three Glasgow hospitals included comprises single-room wards, whilst the other two and the Sheffield site comprise bay wards.

### Derivation of prior probability for post-admission infection

If we assume a uniform individual-level hazard (λ) of infection from 1 February 2020 ($t_0$), whether in hospital or not, then the probability density function (PDF) of infection at time $t_{inf}$ from this date is $\lambda e^{(-\lambda t_{inf})}$. The PDF of infection at time $t_{inf}$ conditional on this occurring at any point prior to the date of symptom onset ($t_{onset}$) is $(\lambda e^{(-\lambda t_{inf})})/(1 - e^{(-\lambda t_{pos})})$, which is approximately $1/t_{onset}$ for small λ (taking the limit as λ->0). For HOCI cases, we are interested in whether $t_{inf}$ occurred before or after the time

of admission to hospital ($t_{adm}$). Also considering the evidence provided by the known incubation time of the disease (PDF $f$ and CDF $F$), we integrate over the range of possible infection dates:

$$
\begin{aligned}
P\big(t_{adm} \leq t_{inf} | t_{inf} \leq t_{onset}, T_{onset} = t_{onset}\big) &= \left[\int_{t_{adm}}^{t_{onset}} f(t_{onset}-x)/t_{onset}.dx\right] / \left[\int_{0}^{t_{onset}} f(t_{onset}-x)/t_{onset}.dx\right] \\
&\approx \left[\int_{t_{adm}}^{t_{onset}} f(t_{onset}-x)/t_{onset}.dx\right] / \left[\int_{-\infty}^{t_{onset}} f(t_{onset}-x)/t_{onset}.dx\right] \\
&= \left[\int_{t_{onset}-t_{adm}}^{0} -f(u)/t_{onset}.du\right] / (1/t_{onset}) \\
&= \int_{0}^{t_{onset}-t_{adm}} f(u).du \\
&= F(t_{onset}-t_{adm})
\end{aligned}
$$

## Geographic weighting for community reference set

### Geographic weighting function

The weight of each sequence within the community reference set is determined by geographic distance from the residential outer postcode of the focus case using a function of the form

$$
\text{weight} = (1-\beta) * \exp(-\tau * \text{community Distance To Index}[i]) + \beta,
$$

where $\beta$ takes a value between 0 and 1, and $\tau > 0$. These parameters are set based on calibration to the available community reference set at each site. The rationale for this weighting is that there is likely to be geographic clustering of viral lineages, and so newly observed community transmissions of SARS-CoV-2 are more likely to show genetic similarity to past sequences from the local area of that individual's home than to past sequences from regions that are further away. If postcode is missing for a case in the community reference set, then distance to the focus sequence is set to 100 km.

### Statistical model for derivation of geographic weighting parameters

The statistical model for geographic weighting is fitted separately for each study site using sequences which are strongly thought to represent community-acquired infection: all community-sampled sequences and patients presenting to A and E with COVID-19, excluding those who are recorded as being HCWss or who do not have an available valid outer postcode. We will refer to these sequences as the 'calibration set'.

A statistical model is constructed to find the optimal values of $\beta$ and $\tau$ to maximise the estimated probability ($P_{sim:i}$) of a newly observed community-acquired case having a similar sequence ($\pm$2 SNPs) to that observed for each sequence in the calibration set. The estimated probability in each case within the calibration set is calculated as a weighted sum of 'close match' indicator variables for all other sequences in the calibration set sample from 6 weeks prior up until the sample date of that case, with the weighting function defined in terms of geographic distance between residential outer postcodes and the $\beta$ and $\tau$ parameters as described for the community reference set.

An overall log-likelihood function is defined using a Bernoulli distribution for each of the $n$ sequences within the calibration set:

$$
\mathfrak{l} = \sum_{i=1}^{n} log(P_{sim:i})
$$

The values of $\beta$ and $\tau$ that maximise $\mathfrak{l}$ were obtained for each of the study sites using the 'bbmle' package for R, with logit-parameterisation of $\beta$ and log-parameterisation of $\tau$.

We assume that the probability of a sequence match conditional on infection from visitor on unit/ward can be calculated using the same weighting scheme as for the probability of a sequence match conditional on community-acquired infection (i.e. P(seq$\pm$2 SNPs|CI)==P(seq$\pm$2 SNPs|VI)).

### Additional matching on ward location history

There is the potential for the algorithm described to return large numbers of close sequences matches with the hospital as a whole, which may make it difficult for IPC teams to use the output to direct their investigations when there are no potential sources of infection identified on the same ward as the focus case. We propose a location matching procedure in order to highlight the most

relevant sequence matches for further investigation. This process does not currently form part of the statistical model, meaning that it can be treated as optional functionality for the SRT in the COG-UK HOCI study, and we have restricted the input data to a simplified format in order to minimise data management requirements.

For each inpatient sample in the input metadata for the algorithm, we specify a single string variable comprising the concatenated names of any ward locations in the ≤14 days prior to the sample date and a separate string variable with any ward locations in the ≤14 days after the sample date. For each focus case submitted to the algorithm, output is flagged if there is any match identified between the wards listed in each of these fields or the ward at time of sampling for a close sequence match in comparison to the prior and current ward locations for the focus sequence (excluding those cases where there is already matching ward location at time of sampling for each).

## Details of phylogenetic methods

Phylogenies were produced by the grapevine pipeline (https://github.com/COG-UK/grapevine) as part of the COG-UK Consortium (https://www.cogconsortium.uk). Briefly, sequences from GISAID and those produced as part of the COG-UK Consortium are independently quality controlled and aligned to the Wuhan reference using minimap2 (https://doi.org/10.1093/bioinformatics/bty191). The two alignments are then combined, the homoplasy at site 11083 is masked and the tree is reconstructed using FastTreeMP (http://www.microbesonline.org/fasttree/). For each of the hospitals of interest, the tree is pruned to keep sequences from Scotland or Yorkshire (as relevant) and by date excluding sequences subsequent to the last 'focus' patient sample date on the ward.

## Details of SRT report format

The SRT system for prospective use needs to provide useful and appropriate feedback in both low-incidence and high-incidence settings for new HOCI cases. This is planned through the generation of a concise one-page PDF summary report for each focus sequence. This summary report contains key focus sequence metadata, information regarding the estimated probabilities for infection source and details of up to 10 close sequence matches identified within the same unit/ward and/or elsewhere in the hospital.

## Probability summary categories

The sequence matching and probability score algorithm generates probability estimates for the source of infection for the focus patient being from the current unit/ward, from elsewhere in the hospital, from the community (pre-admission) or from a visitor. These probability estimates always sum to 1. In the summary report, probability estimates for each source of infection are categorised using the following levels:

- 0–30%: low
- 30–50%: moderately low
- 50–70%: probable
- 70–85%: high
- 85–100%: very high

For clarity of presentation and communication, probability categories will not always be displayed in the summary report for all four potential sources of infection (i.e. ward/unit, elsewhere in hospital, visitor or community). Special handling rules for specific situations are described below.

### Close sequence matches within the same unit and/or hospital

The maximum number of close sequence matches that can be listed on the one-page summary report is 10 (for the combined sum of unit-level and institution-level matches). If the number of ward-level matches is n > 5 and the total number of close sequence matches is N > 10, then the number of ward-level matches is truncated at $5 + \max((5 - (N - n)), 0)$. If there are over 10 close sequence matches in total, then the following message is displayed 'Over 10 close matches; see detailed report for further information'.

Within each set of unit-level and institution-level close sequence matches, ordering and priority for inclusion within the available slots is determined by the following set of criteria (in decreasing order of importance):

1. Number of SNPs relative to Wuhan strain present in comparison sequence but absent in focus sequence (fewer = higher priority)
2. Number of SNPs relative to Wuhan strain present in focus sequence but absent in comparison sequence (fewer = higher priority)
3. Whether comparison sequence is from a HCW (HCWs listed first)
4. HCAI status of comparison sequence (priority order: definite, probable, indeterminate, otherwise)
5. Samples from the past before samples in future
6. Samples from within the 2 weeks prior to focus sequence sample date before others
7. Number of units overlapping with focus sample's units

## Report messages for specific output combinations

### No close sequence matches on unit/ward

If there are no close sequence matches to the focus sequence on their current unit/ward, then no probability category is reported for this potential infection source (the algorithm returns a zero probability in such cases, which could be misleading given uncertainty over screening and sequencing coverage). The message 'No matches from within unit' is displayed. The probability score category for infection from elsewhere in the hospital is provided in such cases.

### No close sequence matches elsewhere in hospital

If there are no close sequence matches to the focus sequence elsewhere in the hospital, then no probability category is reported for this potential infection source. The message 'No matches elsewhere in hospital' is displayed.

### No evidence of transmission within unit or hospital for probable or definite HCAI

If the estimated probability of community-acquired infection from the algorithm is >50%, but the interval from admission to symptom onset (if recorded) or sample date is ≥8 days, then the following message is displayed in place of the estimated probability of community-acquired infection: 'This is a probable/definite HCAI based on admission date, but we have not found genetic evidence of transmission within the hospital'.

### Probable unit- or hospital-acquired infection with source unclear

If the posterior probability of unit-acquired infection and the posterior probability of infection from a source elsewhere in the hospital are each estimated to be <50%, but the sum of these two posterior probabilities is ≥50%, then the following message is displayed: 'Overall, this is a probable unit- or institution-acquired infection with source unclear'.

## Timeline graph

The timeline graph provides a visual representation of available sequences from the same unit/ward and the same institution/hospital as the focus sequence in the period from 3 weeks prior to their sample date to 1 week after. The key indicates which sequences are close matches to the focus sequence, and the numbering corresponds to that in the tabular summary of most relevant close sequence matches.

## Sequencing prioritisation for prospective use of the SRT

The SRT algorithm was initially designed for use with comprehensive sequencing of all SARS-CoV-2 cases within a hospital, in combination with representative sequencing of community-sampled cases.

However, it may be difficult to achieve high population sequencing coverage in some situations, such as if there is a sudden surge in new admissions to the hospital and/or in new HOCI cases. In such scenarios, we have recommended the following prioritisation of samples (from highest to lowest) for sequencing within the prospective HOCI study (https://clinicaltrials.gov/ct2/show/NCT04405934):

1. HOCI cases
2. SARS-CoV-2 +ve patients on wards where there is a HOCI case
3. HCWs with known contact with HOCI cases
4. Other HCWs
5. SARS-CoV-2 +ve patients admitted to any other wards
6. SARS-CoV-2 +ve patients attending for acute care (e.g. Accident and Emergency) but not admitted

These prioritisation rules are guided by the following rationale:

Most probable and definite HCAIs (based on time from admission) show a close sequence match to at least one other case on the same ward, so sequencing of HOCI cases and any cases on the same ward would be enough to identify these links.

- Links between ward outbreaks will be of particular importance to IPC investigations and would be identified with sequencing focused on HOCI cases.
- The probability calculations within the SRT are most important for indeterminate HCAIs, and where there is no sequence match on the same ward the estimated probability of nosocomial infection is <50% in the majority of such cases (36/38 for the Sheffield dataset). The probability estimates for indeterminate HCAIs should be interpreted with caution where overall sequencing coverage is poor, but SRT results are unlikely to lead to inappropriate changes to standard IPC actions if groups '1', '2' and '3' have been sequenced.
- Where there is a complete lack of close sequence matches within the hospital for probable or definite HCAIs, the SRT returns the message that there is a lack of available genetic evidence for linkage (but not that nosocomial infection is unlikely).

Following from this reasoning, we feel that useful information would be returned by the SRT as long as high sequencing coverage is achieved for groups '1', '2' and '3'. High sequencing coverage of groups '4', '5' and '6' would allow the SRT to identify potential links between cases that would likely be missed by standard IPC investigations.

For indeterminate HCAIs with no close sequence matches on the same ward, an inaccurate 'zero' posterior probability of post-admission infection will be returned if one or more similar sequences are found in the community reference set but no similar sequences are observed in the institution reference set with imperfect sequencing coverage. This is likely to be a more important issue in the setting of low SARS-CoV-2 incidence.

For example, if there are 40 cases that could be included in the institution reference set for a focus sequence and 2 of these (5%) would be a close sequence match, then we would need to sequence at least 31/40 (77.5%) in order to have ≥95% probability of observing at least one of the close sequence matches. However, if there are 200 cases that could be included in the institution reference set and 10 of these (5%) would be a close sequence match, then we would need to sequence at least 51/200 (25.5%) in order to have ≥95% probability of observing at least one of the close sequence matches.

A similar relationship would also be observed if we consider a rarer sequence type. If there are 40 cases that could be included in the institution reference set for a focus sequence and 1 of these (2.5%) would be a close sequence match, then we would need to sequence at least 38/40 (95%) in order to have ≥95% probability of observing the one close sequence match. However, if there are 200 cases that could be included in the institution reference set and 5 of these (2.5%) would be a close sequence match, then we would need to sequence at least 90/200 (45%) in order to have ≥95% probability of observing at least one of the close sequence matches.

On this basis, we believe that the goal of close to 100% sequencing coverage should be pursued in the setting of low incidence of SARS-CoV-2, but that overall sequencing coverage of 50% or more may be sufficient in the event that a high incidence of SARS-CoV-2 leads to too great a case load for available sequencing resources.

## Results
### Sequencing coverage in Glasgow dataset

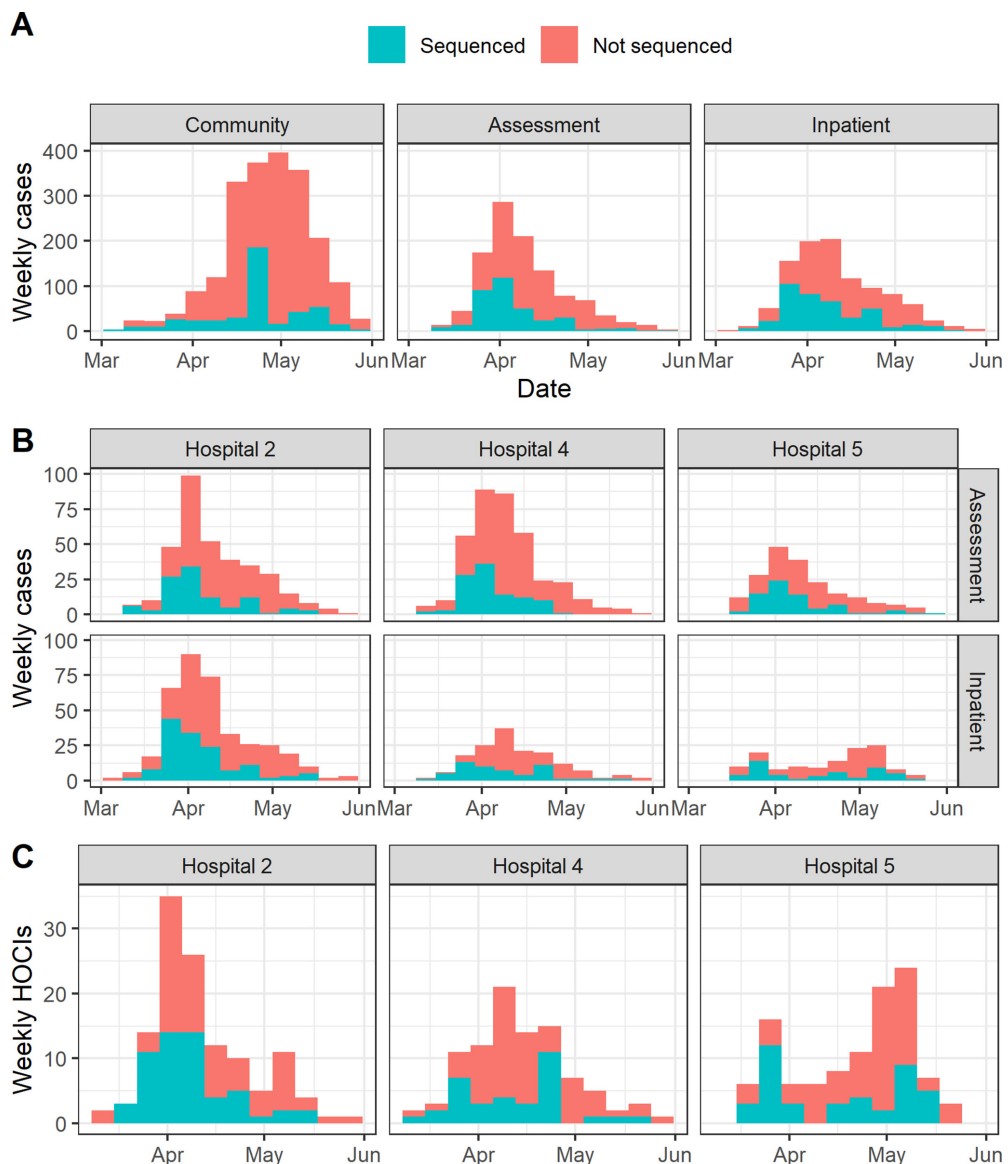

**Appendix 1—figure 1.** Proportion of cases sequenced in Greater Glasgow and Clyde Health Board between 1 March and 27 May (with sequence available as of 23 June 2020) by location of test (A). Also displayed are the proportion of sequenced cases in the three focus hospitals subdivided by assessment and inpatient locations (B), and the proportion of hospital onset COVID-19 infection (HOCI) cases sequenced at these hospitals (C).

## Home residence locations and geographic model parameters

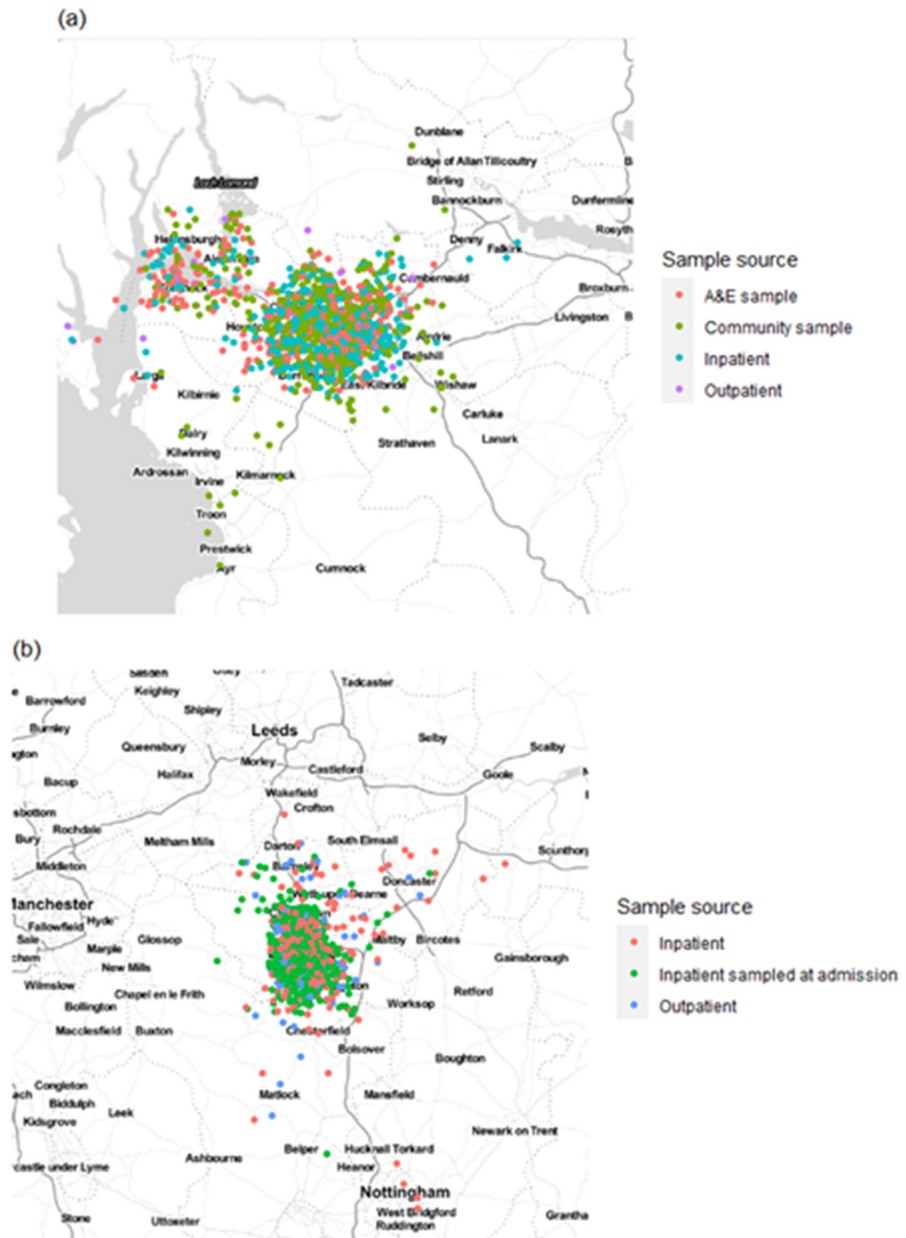

**Appendix 1—figure 2.** Home residence location of individuals in (a) the Glasgow dataset and (b) the Sheffield dataset, displayed by sample source (not including healthcare workers [HCWs]). Locations are analysed using only the outer postcode, and as such random jitter (within longitude and latitude of 0.05) has been added to allow display without overlap of points. Plot created using ggmap for R with map obtained from Stamen maps. For Glasgow, 766 cases were included in the calibration set with estimates of τ = 0.15 and β = 0.0 for the geographic clustering model, whilst for Sheffield 446 cases were included in the calibration set with resulting estimates of τ = 0.84 and β = 0.16.

## SNP distance distributions

For the Glasgow sequence dataset as a whole, the median pairwise SNP difference among all sequences was 9, and there were 1.3, 3.4, 6.4 and 10.1% of pairwise comparisons with 0, ≤1, ≤2

and $\leq$3 SNP differences, respectively. For the Sheffield dataset as a whole, the median pairwise SNP difference among all sequences was 8, and there were 1.2, 3.3, 6.5 and 10.8% of pairwise comparisons with 0, $\leq$1, $\leq$2 and $\leq$3 SNP differences, respectively.

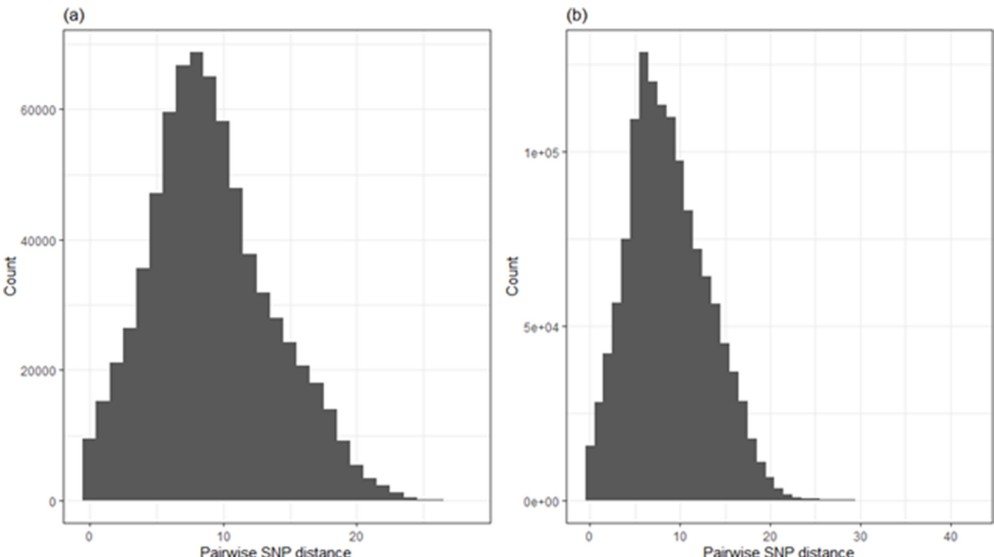

**Appendix 1—figure 3.** Frequency plot of all pairwise SNP differences among (**a**) all 1199 sequences in the Glasgow dataset and (**b**) all 1629 analysed sequences in the Sheffield dataset.

## Additional case study

*Appendix 1—figure 4* shows a phylogenetic tree indicating complex transmission networks across multiple hospitals in the Glasgow area (with SRT outputs for Hospitals 2 and 4). A monophyletic cluster of HOCIs can be seen in Hospital 2 Unit 48, with the first detected case identified by the SRT as a hospital-acquired and the rest unit-acquired infections. A paraphyletic group of HOCIs was detected in Hospital 4 Unit 69. Patient 1 (UID0042) was screened for COVID in Unit 69 on 14 April 2020 after developing a cough and oxygen requirement. The patient was moved from the nightingale area to a single room on the ward on 14 April 2020 and was confirmed positive on 15 April 2020.

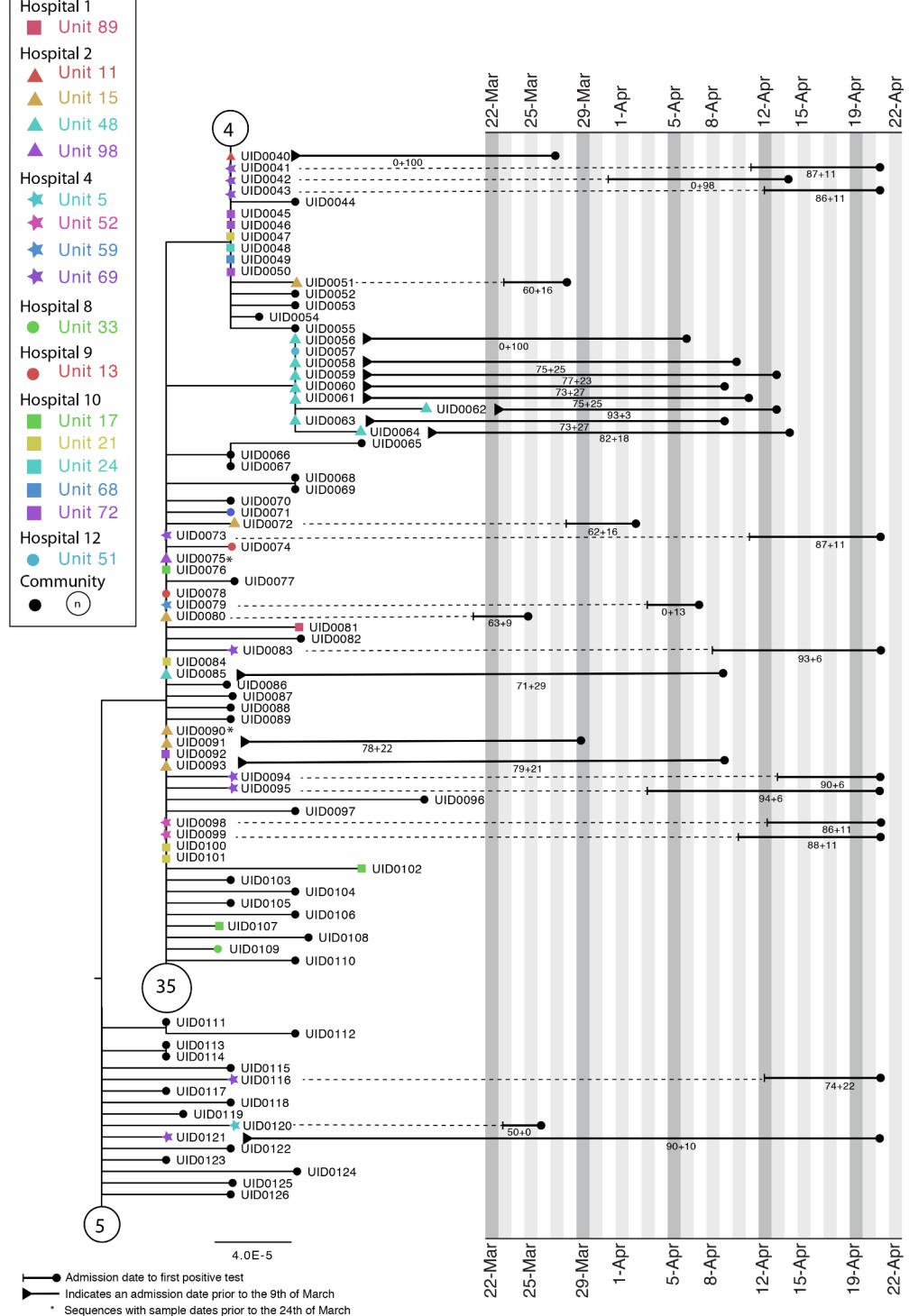

**Appendix 1—figure 4.** Maximum-likelihood tree for sequences found in Hospital 2 Unit 48 and Hospital 4 Unit 69 of the Glasgow dataset up until the 21st of April (inclusive). The circles with numbers represent the number of community sequences that are identical and at the base of each lineage (n = 5, n = 35, n = 4). Tree tips with black circles represent further community sequences. The black lines represent the time from admission to sampling. The values below the line are the posterior probability for unit infection + the posterior probability of hospital infection from the sequence reporting tool.

On 20 April 20, a second patient on Unit 69 (not sequenced) was screened after developing a cough and pyrexia and confirmed positive on 21 April 2020. The patient was in a single room at the time of symptom onset; however, they had been in the main nightingale ward opposite patient 1 for 5 days. At this point 13 asymptomatic contacts in Unit 69 were screened, and 8 (UID0043, UID0073, UID0041, UID0095, UID0116, UID0094, UID0083, UID0121) were positive. These cases are all identified as hospital-acquired or unit-acquired infections and can be grouped into a genetically similar cluster with a maximum pairwise distance of two SNPs between each member and its nearest neighbour. However, this cluster clearly represents multiple introductions of SARS-CoV-2 onto the ward.

Examples of SRT reports

**Appendix 1—figure 5.** Example of sequence reporting tool output with estimated very highly probable infection within unit.

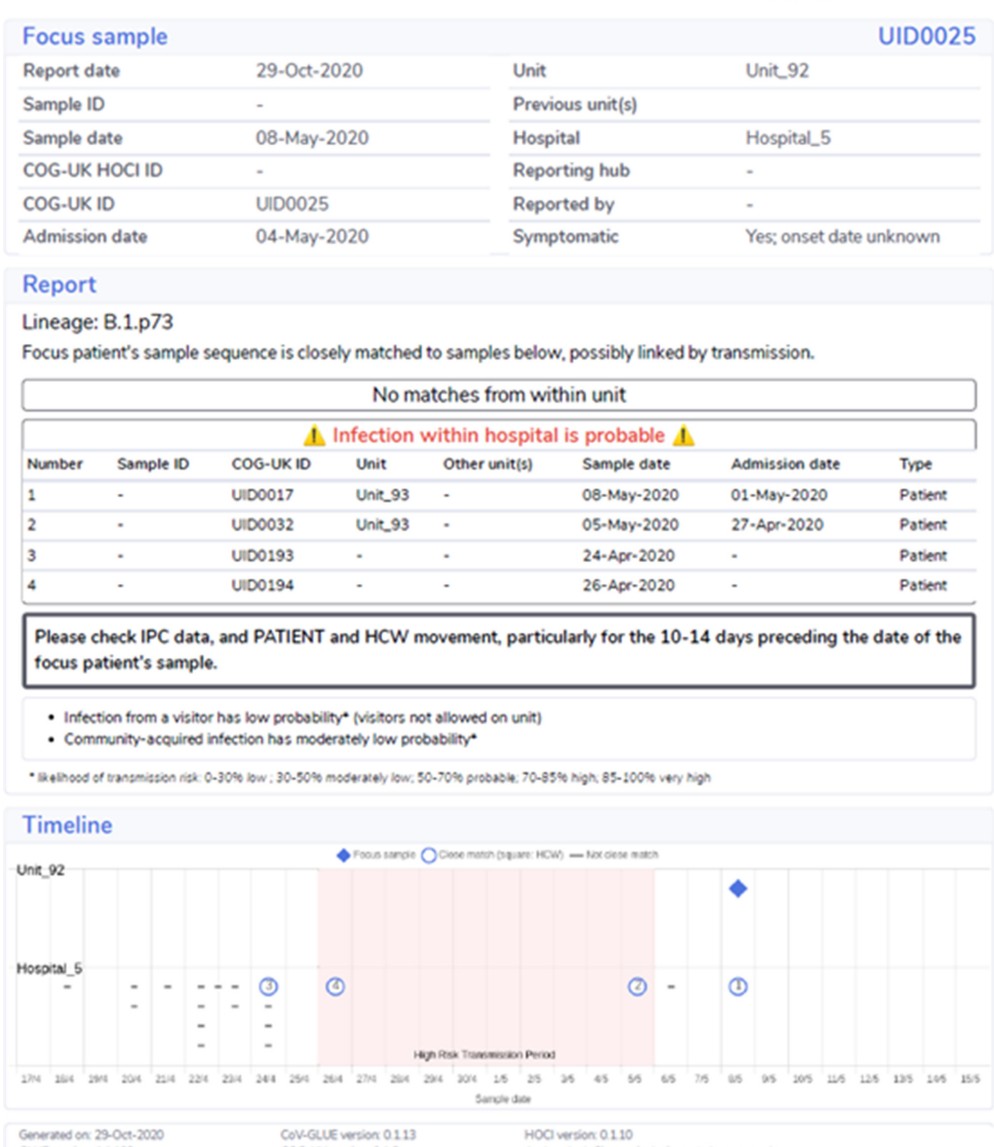

**Appendix 1—figure 6.** Example of sequence reporting tool output with estimated probable infection within hospital.

