## [Decision Letter]

**Acceptance summary:**

This paper by Stirrup and colleagues describes a methodology that can be useful for for real-time investigation of COVID-19 outbreaks and SARS-CoV-2 infection prevention by infectious disease control teams in the hospital setting. It combines epidemiological and viral sequencing data – rather than only epidemiological data – to identify and investigate the infection source of hospital onset SARS-CoV-2 infections. The challenge for the use of this methodology is to secure the timely availability of the sequence data and requires close collaboration between methodologists, virologists, infectious disease clinicians and software engineers, to collectively create the appropriate workflows and reporting systems.

**Decision letter after peer review:**

Thank you for submitting your article "Rapid feedback on hospital onset SARS-CoV-2 infections combining epidemiological and sequencing data" for consideration by *eLife*. Your article has been reviewed by 2 peer reviewers, including Albert Osterhaus as the Reviewing Editor and Reviewer #1, and the evaluation has been overseen by Jos van der Meer as the Senior Editor.

Essential revisions:

The method was validated retrospectively with data from two UK city hospital settings and is eventually meant to be used for future use for the identification, prevention and control of healthcare-associated SARS-CoV-2 infections in hospitals.

Indeed, there is an urgent need for tools that can synthesise multiple data streams to provide real time information to healthcare professionals. It is questionable to what extent the tool presented is generalisable to medical facilities outside of the specific data rich settings considered here, or if the tool is useful for prospective analyses.

Concerning the question to what extent the tool presented is generalisable to medical facilities. the papers' claims are largely supported by the data and may be useful for SARS-CoV-2 infection prevention and control teams, provided they effectively have access to, and can implement both elements of the system: evaluation of state-of-the art epidemiological data and of phylogenetic sequence data based on local and hospital SARS-CoV-2 genomes as determined within 24-48 hours by the COVID-19 Genomics (COG) UK initiative. Therefore this study would rather be of interest to specialists working in hospital infection prevention, with more limited further interest.

Concerning the prospective analysis of prospective use cases described in the discussion and given the high burden of SAR-CoV-2 in the UK in late 2020, there should already be ample data available for their inclusion.

As the Discussion section briefly touches on the motivation for using a 2 SNP threshold for identifying "similar" viruses instead of phylogeny-informed approaches, it dismisses the latter saying that they are too intensive to be used outside of research-intensive settings. Although phylogenetic approaches would be more resource intensive, it would seem that the amount of sequencing and shortness of turnaround time required to make this algorithm useful similarly limit its utility to research-intensive settings. With this in mind, the substantially increased information associated with phylogenetic approaches would seem to be worth the increased resource costs particularly since the capabilities and quality of phylogenetic tools has increased substantially since the publication of the 2013 paper cited on line 326. Although extending this particular algorithm to include phylogenetic data is outside the scope of the study, the manuscript is unfairly dismissive of such approaches, especially when a location specific build of nextstrain can be created with a programming burden similar to that of the algorithm described in this study.

Given these considerations, the paper would be acceptable for publication if the following points would be addressed:

1) Analysis of the prospective use cases described in the discussion. This would seem essential for validating the utility of the algorithm.

2) Address specifically the limitations of the study that are related to the overall applicability of the methodology in settings with more limited access to state-of-the art epidemiological and phylogenetic sequence data, the latter based on timely availability of local and hospital SARS-CoV-2 genomes.

3) Although extending this particular algorithm to include phylogenetic data is outside the scope of this study, it should address the value of such approaches, especially when a location specific build of nextstrain can be created with a programming burden similar to that of the algorithm described in this study.

*Reviewer #1 (Recommendations for the authors):*

The science is based on a practical comparison of the existing PHE system and the newly established SRT, which is based on epidemiological and digested viral sequenced data. The criteria chosen are largely arbitrary but justified by practical considerations. The presentation of data and conclusions is clear. More attention could be paid to the overall applicability of the methodology (with or without modifications) in other geographical or demographic settings with different public health infrastructures and their limitations.

Tables, figures and supplementary material are well designed and informative

*Reviewer #3 (Recommendations for the authors):*

Foremost, this study could be substantially strengthened through the analysis of the prospective use cases described in the discussion. Given the high burden of SAR-CoV-2 in the UK in late 2020, there should already be ample data available for inclusion. This would seem essential for validating the utility of the algorithm.

It would also be very helpful to include some estimates on the circumstances under which the algorithm is likely to provide useful information. The UK is highly unusual in terms of its virus sequencing efforts. What proportion of test-positive cases need to be sequenced in order for the algorithm to provide reliable estimates of ongoing transmission under different outbreak scenarios? On a related subject, the impact of turnaround time from sample collection to receipt of sequence data should be evaluated. How fast does sequence data need to be able available for the algorithm to provide actionable information?

The Discussion section briefly touches on the motivation for using a 2 SNP threshold for identifying "similar" viruses instead of phylogeny-informed approaches but dismisses the latter saying that they are too intensive to be used outside of research-intensive settings. I agree that phylogenetic approaches would be more resource intensive, but it would seem that the amount of sequencing and shortness of turnaround time required to make this algorithm useful similarly limit its utility to research-intensive settings. With this in mind, the substantially increased information associated with phylogenetic approaches would seem to be worth the increased resource costs particularly since the capabilities and quality of phylogenetic tools has increased substantially since the publication of the 2013 paper cited on line 326. I recognise that extending this particular algorithm to include phylogenetic data is outside the scope of work for this project, but the manuscript is unfairly dismissive of such approaches, especially when a location specific build of nextstrain can created with a programming burden similar to that of the algorithm described in this study.

---

## [Author Response]

Essential revisions:The method was validated retrospectively with data from two UK city hospital settings and is eventually meant to be used for future use for the identification, prevention and control of healthcare-associated SARS-CoV-2 infections in hospitals.Indeed, there is an urgent need for tools that can synthesise multiple data streams to provide real time information to healthcare professionals. It is questionable to what extent the tool presented is generalisable to medical facilities outside of the specific data rich settings considered here, or if the tool is useful for prospective analyses.Concerning the question to what extent the tool presented is generalisable to medical facilities. the papers' claims are largely supported by the data and may be useful for SARS-CoV-2 infection prevention and control teams, provided they effectively have access to, and can implement both elements of the system: evaluation of state-of-the art epidemiological data and of phylogenetic sequence data based on local and hospital SARS-CoV-2 genomes as determined within 24-48 hours by the COVID-19 Genomics (COG) UK initiative. Therefore this study would rather be of interest to specialists working in hospital infection prevention, with more limited further interest.Concerning the prospective analysis of prospective use cases described in the discussion and given the high burden of SAR-CoV-2 in the UK in late 2020, there should already be ample data available for their inclusion.

We agree that there remain open questions regarding the feasibility and utility of routine near real-time pathogen sequencing for Infection Prevention and Control (IPC), and regarding the factors that might affect this in different healthcare settings. As noted in our cover letter for this revised manuscript, we are in the process of undertaking a large-scale prospective evaluation of viral sequencing and use of the tool for SARS-CoV-2 IPC as part of a broader research program that will include measurement of its impact on the knowledge and actions of IPC teams. The prospective trial includes 14 National Health Service Trusts from across the UK, representing a broad range of settings – from university hospitals to a district general hospital – and the research outputs will include quantitative, qualitative and health economic evaluations of impact and feasibility.

We believe that automated tools to provide feedback on genetic and epidemiological data are needed in order to maximise the feasibility and utility of routine pathogen sequencing across a range of healthcare settings.

As the Discussion section briefly touches on the motivation for using a 2 SNP threshold for identifying "similar" viruses instead of phylogeny-informed approaches, it dismisses the latter saying that they are too intensive to be used outside of research-intensive settings. Although phylogenetic approaches would be more resource intensive, it would seem that the amount of sequencing and shortness of turnaround time required to make this algorithm useful similarly limit its utility to research-intensive settings. With this in mind, the substantially increased information associated with phylogenetic approaches would seem to be worth the increased resource costs particularly since the capabilities and quality of phylogenetic tools has increased substantially since the publication of the 2013 paper cited on line 326. Although extending this particular algorithm to include phylogenetic data is outside the scope of the study, the manuscript is unfairly dismissive of such approaches, especially when a location specific build of nextstrain can be created with a programming burden similar to that of the algorithm described in this study.

Thank you for these comments, we have adjusted the text of the Discussion to better reflect the nuances raised regarding computational and staff resource demands for phylogenetic investigation in comparison to our approach.

Despite substantial advancements in phylogenetic methodology and software since 2013, we feel that the statement “While a phylogenetic approach is useful in excluding direct transmission between cases, it can be more problematic to confirm transmission source” remains true. For example HIV has a much higher mutation rate than SARS-CoV-2 which allows for use of within-host viral diversity as an additional source of information for inference regarding ‘who infected whom’, but the level of evidence obtained is still often inconclusive at the individual level (e.g. as reported by Ratmann et al., doi.org/10.1038/s41467-019-09139-4).

Given these considerations, the paper would be acceptable for publication if the following points would be addressed:1) Analysis of the prospective use cases described in the discussion. This would seem essential for validating the utility of the algorithm.

As discussed with the Editors and noted in the cover letter, we have not included additional prospectively collected data in the resubmitted manuscript. We agree that there is a need for robust prospective evaluation of viral sequencing for IPC and of this tool specifically. However, the present work constitutes a distinct element within a broader project, which we believe will form an important body of work regarding the routine use of viral sequencing for IPC. This has been clarified in a new paragraph in the Discussion:

“Prospective evaluation of the SRT is currently underway within a multicentre study in the UK^[32]^[Protocol REF]. This study and its accompanying research program will evaluate the impact of routine viral sequencing and use of the SRT on IPC knowledge, actions and outcomes, and will include quantitative, qualitative^[33]^ and health economic analyses to help guide the future development of pathogen genomics for IPC.”

2) Address specifically the limitations of the study that are related to the overall applicability of the methodology in settings with more limited access to state-of-the art epidemiological and phylogenetic sequence data, the latter based on timely availability of local and hospital SARS-CoV-2 genomes.

We have expanded on these issues as suggested in the Discussion:

“The automated feedback provided by the SRT is nonetheless dependent on timely sequencing of a high proportion of viral samples from cases within the hospital concerned, ideally in combination with sequences also available from community-sampled cases. […] This judgement also would be dependent on the available health infrastructure and resources at both the local and national levels.”

3) Although extending this particular algorithm to include phylogenetic data is outside the scope of this study, it should address the value of such approaches, especially when a location specific build of nextstrain can be created with a programming burden similar to that of the algorithm described in this study.

We have adjusted the text of the Discussion to better reflect the value of phylogenetic investigation as suggested:

“There have been advances in recent years in the computational efficiency and workflow standardisation possible for phylogenetic analyses that have made it easier to use these methods for real-time investigation of outbreaks, for example through the development of the Nextstrain project^[28, 29]^. […] There will be cases in which phylogenetic analysis would provide information beyond that returned by the SRT, and the two approaches may be complementary to one another for outbreak investigation.”

The following text has been deleted from the Discussion:

“However, fully integrated epidemiological and phylogenetic analysis of hospital outbreaks is resource-intensive, presenting challenges in delivering the rapid turnaround and scale-up required to provide clear feedback to hospital IPC teams outside of research-intensive settings.”

Reviewer #1 (Recommendations for the authors):The science is based on a practical comparison of the existing PHE system and the newly established SRT, which is based on epidemiological and digested viral sequenced data. The criteria chosen are largely arbitrary but justified by practical considerations. The presentation of data and conclusions is clear. More attention could be paid to the overall applicability of the methodology (with or without modifications) in other geographical or demographic settings with different public health infrastructures and their limitations.Tables, figures and supplementary material are well designed and informative.

We thank the reviewer for their comments on our work. As suggested, we have expanded the Discussion in relation to the broader applicability and limitations of the methodology developed:

“…further evidence is required to determine whether rapid sequencing is worth the necessary investment for routine use within IPC practice. This judgement also would be dependent on the available health infrastructure and resources at both the local and national levels.”

Reviewer #3 (Recommendations for the authors):Foremost, this study could be substantially strengthened through the analysis of the prospective use cases described in the discussion. Given the high burden of SAR-CoV-2 in the UK in late 2020, there should already be ample data available for inclusion. This would seem essential for validating the utility of the algorithm.It would also be very helpful to include some estimates on the circumstances under which the algorithm is likely to provide useful information. The UK is highly unusual in terms of its virus sequencing efforts. What proportion of test-positive cases need to be sequenced in order for the algorithm to provide reliable estimates of ongoing transmission under different outbreak scenarios?

The algorithm was initially designed with the aim of close to 100% sequencing coverage of positive cases within the hospital. This reflects a potential future in which pathogen sequencing is a routine part of lab diagnostics, but as suggested this can present a substantial challenge at present – and one which is highly dependent on the local incidence of new cases. The retrospective evaluation that we have conducted indicates that performance may be acceptable with overall sequencing coverage at substantially lower levels (34% of known cases for Glasgow and 73% for Sheffield). However, the performance will also be influenced by which patients are being prioritised for sequencing and the incidence rates of admission of COVID-19 patients and of the occurrence of HOCI cases.

We have added a new section to the Appendices regarding this issue ‘Sequencing prioritisation for prospective use of the SRT’, and have included the sequencing prioritisation guidelines and associated rationale that have been developed in the course of running our prospective study.

On a related subject, the impact of turnaround time from sample collection to receipt of sequence data should be evaluated. How fast does sequence data need to be able available for the algorithm to provide actionable information?

We feel that this question requires prospective evaluation. As noted in our response to the Public Review, our ongoing prospective study includes intervention phases both with a ‘rapid’ target turnaround of 48 hours from sampling and with a ‘slow’ target turnaround of 5-10 days, and this will generate data on the relative utility for IPC of viral sequencing within these timeframes.

The Discussion section briefly touches on the motivation for using a 2 SNP threshold for identifying "similar" viruses instead of phylogeny-informed approaches but dismisses the latter saying that they are too intensive to be used outside of research-intensive settings. I agree that phylogenetic approaches would be more resource intensive, but it would seem that the amount of sequencing and shortness of turnaround time required to make this algorithm useful similarly limit its utility to research-intensive settings. With this in mind, the substantially increased information associated with phylogenetic approaches would seem to be worth the increased resource costs particularly since the capabilities and quality of phylogenetic tools has increased substantially since the publication of the 2013 paper cited on line 326. I recognise that extending this particular algorithm to include phylogenetic data is outside the scope of work for this project, but the manuscript is unfairly dismissive of such approaches, especially when a location specific build of nextstrain can created with a programming burden similar to that of the algorithm described in this study.

We have adjusted the Discussion to better reflect the comparative advantages and limitations with respect to phylogenetic investigation:

“There have been advances in recent years in the computational efficiency and workflow standardisation possible for phylogenetic analyses that have made it easier to use these methods for real-time investigation of outbreaks, for example through the development of the Nextstrain project^[28, 29]^. […] There will be cases in which phylogenetic analysis would provide information beyond that returned by the SRT, and the two approaches may be complementary to one another for outbreak investigation.”